# Learning Dynamic Stability Landscapes in Synchronization Networks

**Christian Nauck** [1]  **Junyou Zhu** [1 2]  **Michael Lindner** [3]  **Frank Hellmann** [1]

## Abstract

The robustness of synchronization is typically characterized by scalar, per-node stability indices whose dependence on topology is studied via network science or graph neural networks (GNNs). We propose a novel upstream task, learning *stability landscapes*, which provide deeper insights into synchronization behavior and from which many such scalar indices can be derived. Crucially, we pioneer a graph-to-image prediction paradigm: learning image-like landscapes as per-node targets directly from graph topology, a formulation we are not aware of having been established elsewhere in the literature. To support this task, we release two datasets of 10,000 graphs each at 20 and 100 nodes with per-node landscape labels, based on a conceptual oscillator model, capturing power grid synchronization behavior. A GNN encodes topology and a CNN decoder renders per-node images, learned end-to-end with good in-distribution accuracy, generalizing across graph sizes and to realistic power grid topologies. This demonstrates that stability landscapes, while beyond the reach of conventional network science, are learnable from topology and open new avenues for moving beyond scalar stability indices in biology, neuroscience, and power grids.

## 1. Introduction

Networks of coupled oscillators play a fundamental role in modeling both natural and engineered systems. They provide a unifying framework for understanding complex dynamics across fields, such as biology, neuroscience, ecology, physics, and engineering. Systems including the heart, the brain, food webs, coupled lasers, chemical reactions, power grids, and even firefly populations can all be described as oscillators interacting on complex networks (Strogatz, 2000; Acebron et al., 2000; Pikovsky et al., 2001; Acebrón et al., 2005; Pecora et al., 2014; Rodrigues et al., 2016). The behavior of these systems is strongly influenced by the underlying network topology which governs who interacts with whom and how perturbations spread.

An important phenomenon in oscillator networks is synchronization. Whether synchronization is desirable depends on the application. In the brain, excessive synchronization can signal dysfunction, as in epilepsy. In power grids, synchronization is essential for stable operation. This duality makes the robustness of the synchronous state a central question in networked complex systems. Understanding and controlling synchronization has practical implications that range from disrupting pathological synchronization in the brain (Tass, 2007) to designing infrastructure with favorable stability properties (Yamamoto et al., 2023; Menck et al., 2013; 2014; Berner et al., 2021).

The resulting core question is whether synchrony remains stable under perturbations, especially local perturbations applied at a specific node $i$. Before introducing the technical framework, we illustrate the motivation using the stability landscape plots in Figure 1. The axes represent perturbation magnitudes: points near the origin correspond to small perturbations, while points farther away correspond to larger perturbations. Dark purple indicates stable outcomes, whereas lighter green and yellow indicate instability. Such landscape visualizations are common in complex-systems research, but generating them is computationally expensive. In this work, we show that these landscapes can be predicted accurately using machine learning.

Formally, consider a dynamical system on a graph $\mathcal{G} = (\mathcal{V}, \mathcal{E}, \mathbf{X})$ with vertices $\mathcal{V}$, edges $\mathcal{E}$ and dynamical parameters $\mathcal{X}$. We focus on basin stability (Menck et al., 2013), where perturbations are modeled as displacements of the variables associated to a node by a perturbation parameters $p$, and a perturbation is labeled as stable if the system returns to synchrony from this initial condition as time goes to infinity. For a node $i \in \mathcal{V}$, the corresponding single-node

---

[1]Department of Complexity Science, Potsdam Institute for Climate Impact Research, Potsdam, Germany [2]Machine Learning Group, Technical University of Berlin, 10587 Berlin, Germany [3]Department of Digital Transformation in Energy Systems, Institute of Energy Technology, Technical University of Berlin, Germany. Correspondence to: Junyou Zhu <junyou.zhu@pik-potsdam.de>, All authors email <nauck@pik-potsdam.de, hellmann@pik-potsdam.de>.

*Proceedings of the 43rd International Conference on Machine Learning*, Seoul, South Korea. PMLR 306, 2026. Copyright 2026 by the author(s).

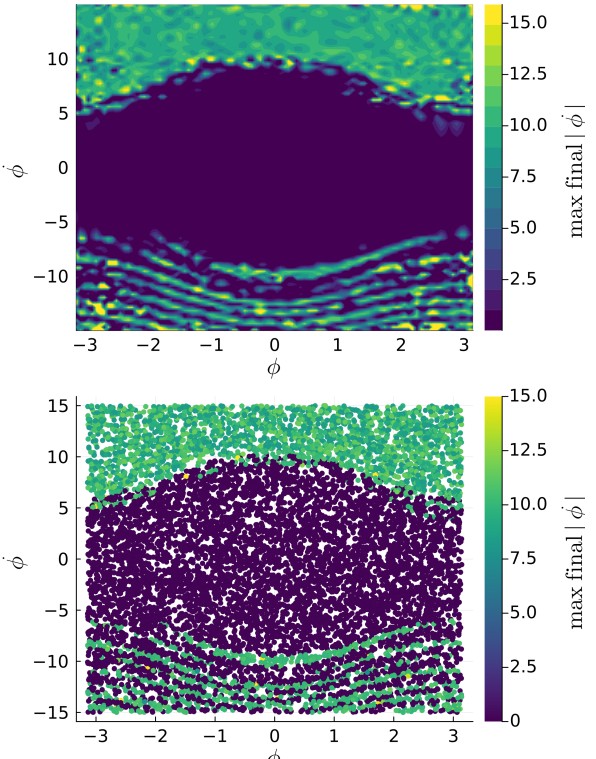

*Figure 1.* Top: Contour plot of the asymptotic deviation from synchrony. Bottom: its Monte Carlo origin landscape for 10,000 perturbations at the same node $i$. The axes represent the perturbations in $p = \{\phi, \dot\phi\}$. The color encodes the maximum final frequency deviation from the set point, with darker colors indicating more stable regions.

basin stability (SNBS) is defined as

$$\text{SNBS}_i(\mathcal{G}) = \int \text{BS}(\mathcal{G}, i, p)\, \rho(p)\, dp, \qquad (1)$$

where $\text{BS}(\mathcal{G}, i, p) \in \{0, 1\}$ indicates whether the synchronous state survives perturbation $p$ at node $i$, and $\rho(p)$ denotes the distribution from which perturbations are sampled over the full perturbation space. Thus, every node $i$ is labeled by the probability $\text{SNBS}_i$ that the system returns to synchrony under a randomly drawn perturbation.

Estimating $\text{SNBS}_i$ using Monte-Carlo integration to a given accuracy requires a fixed number of evaluations of BS per node, so labeling the whole graph thus scales as $|\mathcal{V}||\mathcal{E}|$ in the best case. For conceptual models of power grids, this labeling has become an important tool for understanding the impact of topological features on overall stability properties (Witthaut et al., 2022; Kim et al., 2016; Nitzbon et al., 2017). Further it was shown in (Nauck et al., 2022b;a; 2023; Zhu et al., 2026) that these labels can be well predicted by GNN methods. Not only are these GNN evaluation dramatically faster than evaluating $\text{SNBS}_i$ by sampling, they also scale

as $|\mathcal{E}|$ with graph size.

A disadvantage of predicting only $\text{SNBS}_i$ is that it requires choosing a meaningful perturbation distribution $\rho(p)$ when generating the dataset. Different situations of practical interest, and different stability aspects, are captured by different choices of $\rho$, or by more complex expectation values $\int O(p)\, \text{BS}(\mathcal{G}, i, p)\rho(p)\, dp$ of some observable $O$. This motivates us to introduce in this paper a new upstream task for stability prediction.

Instead of averaging over the entire perturbation space at once, we decompose it into regions of similar perturbations. Let $h$ denote one such region, and let $\rho(p \mid h)$ be a distribution concentrated on region $h$, that is, sampling perturbations only within that part of the perturbation space. In the concrete setting studied here, $p = (\phi, \dot\phi)$ lies in the phase–frequency plane, $\rho(p)$ corresponds to sampling perturbations over the full perturbation space, and $\rho(p \mid h)$ corresponds to sampling perturbations only within the region or grid cell $h$. We then define the basin stability landscape (LBS) for node $i$ as

$$\text{LBS}_i(\mathcal{G}, h) = \int \text{BS}(\mathcal{G}, i, p)\, \rho(p \mid h)\, dp. \qquad (2)$$

Intuitively, $\text{LBS}_i(\mathcal{G}, \cdot)$ can be visualized as a stability landscape, that is, a heatmap over the perturbation space that highlights safe and unsafe regions for node $i$. Specifically, Figure 1 (a) shows a contour rendering of such a landscape for one node.

Most importantly, the scalar score is recovered as a mixture of regional probabilities:

$$\text{SNBS}_i(\mathcal{G}) = \int \rho(h)\, \text{LBS}_i(\mathcal{G}, h)\, dh. \qquad (3)$$

where $\rho(h)$ denotes the weight of region $h$ in the overall perturbation distribution. Hence, the landscape is a strictly richer object than the scalar basin stability score. In this context, plotting $\text{BS}(\mathcal{G}, i, p)$ as a function of $p$ is typically done for illustrative purposes (Menck et al., 2014; Hellmann et al., 2020; Zhang et al., 2024).

As we are typically only interested in perturbations up to a maximum size, this leads naturally to a finite set of regions. Thus $\text{LBS}_i(\mathcal{G}, h)$ provides a labeling of the graph where every node is labeled by a vector. In the concrete model we will study below, the perturbation vector is two-dimensional, and it is natural to discretize it into boxes, leading to cell-wise labels $\text{LBS}_{i,m,n} \in [0, 1]$. In this work we extend the publicly available $\text{SNBS}_i$ datasets for ensembles of oscillator networks (Nauck et al., 2022a; 2023) by these basin landscapes.

This dataset includes two graph ensembles, 10,000 graphs

with 20 nodes each and 10,000 graphs with 100 nodes each, to provide a valuable testbed for evaluating models' generalization capabilities. Specifically, the availability of two distinct graph scales allows researchers to effectively assess out-of-distribution (OOD) generalization performance across graph sizes, which is especially important given that labeling a full graph scales at least as $|\mathcal{V}||\mathcal{E}|$. Consequently, an ML model capable of generalizing from smaller (e.g., 20-node graph) to larger networks(e.g., 100-node graph) could provide the only practically viable route towards applying complex stability measures in practice.

The main contributions of this work are as follows:

- We introduce a novel stability assessment task by predicting $\text{LBS}_i(\mathcal{G}, h)$ for the first time, which captures advantages of probabilistic approaches, while also enabling varied downstream tasks.

- We present new datasets to support research on this task. The datasets consist of two ensembles, each containing 10,000 graphs with basin landscapes as prediction targets. They were generated using computationally expensive simulations requiring roughly **500,000 CPU hours**.

- As initial baselines, we propose graph neural networks (GNNs) as encoders, with multilayer perceptrons (MLPs) or convolutional neural networks (CNNs) as decoders, demonstrating the potential of machine learning methods to address this critical challenge in networked dynamical systems.

- We establish the first practical benchmark for stability landscape prediction. Our models reach about 85% Structural similarity index measure (SSIM) in distribution and up to 78 % SSIM under a size shift from 20 to 100 nodes. They transfer in a zero-shot setting to four real power grids with up to 80% SSIM, and match Monte Carlo fidelity while cutting evaluation from thousands of CPU hours across the datasets to seconds per graph.

- The new setup pioneers a graph-to-image prediction paradigm: learning image-like landscapes as per-node targets directly from graph topology. To the best of our knowledge, this formulation has not been established elsewhere in the literature, and may prove useful beyond synchronization, wherever a spatial or visual description of node behavior is more informative than single scalars.

## 2. Generation of datasets with basin landscapes

In this section, we present details on how to generate a dataset suitable for the novel supervised ML task. We begin by introducing the underlying dynamical system (the second-order Kuramoto oscillator) and explain the methodology for generating basin stability landscapes via perturbations (see Section 2.1). We then outline how these basin landscapes are converted into heatmaps that are regarded as prediction targets for supervised ML (see Figure 1 (b)). Finally, we summarize the key statistical properties of our dataset (see Section 2.2).

### 2.1. Basin landscapes of dynamical systems

The dynamical systems studied in this work are coupled networks of Kuramoto oscillators. We model each node of these networks as a paradigmatic second-order Kuramoto model (Kuramoto, 1984; 1975):

$$\ddot{\phi}_i = P_i - \alpha\dot{\phi}_i - \sum_{j=1}^{n} KA_{ij}\sin(\phi_i - \phi_j), \qquad (4)$$

where $\phi_i$ is the phase angle at node $i$, $\dot{\phi}_i$, and $\ddot{\phi}_i$ are its first and second time derivatives, respectively. The network topology is encoded in the adjacency matrix $\mathbf{A}$. For all ensembles we fix the damping coefficient to $\alpha = 0.1$ and assume homogeneous coupling $K = 9$. The only node-dependent parameter is the injected power $P_i \in \{-1, +1\}$, which is regarded as the node feature in our ML settings. All dynamical simulations are conducted with respect to a reference frequency. Negative frequencies indicate that the frequency is below the reference frequency, e.g., 50 Hz in the case of the European power grid.

To perform the dynamical simulations, the statically stable state is perturbed using perturbations uniformly sampled in the phase-frequency space $(\phi, \dot{\phi}) \in [-\pi, \pi] \times [-15, 15]$. We apply 10,000 independent perturbations per node, with the results visualized as basin landscapes (see Figure 1 (b)). Each point represents the outcome of one dynamical simulation, and the plot shows the maximum absolute value of the final frequency at all nodes. Darker points (small values of *max final* $|\dot{\phi}|$ ) indicate stable outcomes, whereas lighter points indicate unstable.

Since real applications usually care about whether a given combination of parameter configurations leads to stable or unstable states, rather than the exact value of $|\dot{\phi}|$, we classify simulation outcomes (i.e., points) as either stable or unstable. Following prior work (Nauck et al., 2022b), we adopt the threshold $|\dot{\phi}| \leq 0.1$, considering configurations below this value to be synchronized (dynamically stable).

On this basis, the landscape of each node is represented by 10,000 points, each point being labeled as either stable or unstable. The overall stability probability, known as single-node basin stability (SNBS) (Menck et al., 2013), that a node returns to an overall stable state is defined as the ratio of stable points to the total number of points (i.e., the total number of perturbations applied). For the example in Figure 1 (b), this SNBS value equals the number of dark-purple points divided by the total of 10,000 perturbations applied to the node. Unlike previous univariate node-level regression-based studies (Nauck et al., 2022b;a; 2023), which predict only this scalar SNBS value for each node, we aim to reconstruct the full landscape directly from network topology.

Directly feeding raw basin landscapes as labels into a learning algorithm is not feasible. Each node is perturbed $10,000$ times at randomly chosen phases (i.e., frequency pairs), resulting in a point cloud that differs from node to node in both location and density. Without a common "input shape", standard regression losses or neural network decoders cannot be applied.

Let a single perturbation be denoted by $\boldsymbol{\delta} = (\phi, \dot{\phi}) \in [-\pi, \pi] \times [-15, 15]$, and let the set of all perturbations applied to one node be $\mathcal{D} = \{\boldsymbol{\delta}_1, \ldots, \boldsymbol{\delta}_{10,000}\}$. To address this, we down-sample each landscape onto a regular $20 \times 20$ grid. $20 \times 20$ grids have the highest practical resolution with acceptable sampling noise. Other resolutions are investigated in Appendix A.10.

Denote by $\mathcal{C}_{i,m,n} \subset \mathcal{D}$ the subset of perturbations that fall into grid cell $(m, n)$ for the designated node $i$, and let $q_{i,m,n} = |\mathcal{C}_{i,m,n}|$. We define the empirical *stability probability* in that cell as

$$\text{LBS}_{i,m,n} = \frac{\#\left\{\boldsymbol{\delta} \in \mathcal{C}_{i,m,n} : \max_{j \in \mathcal{V}} \left|\dot{\phi}_j^{\text{final}}(\boldsymbol{\delta})\right| \leq 0.1\right\}}{q_{i,m,n}}$$
$$\in [0, 1]. \tag{5}$$

Here, $\dot{\phi}_j^{\text{final}}(\boldsymbol{\delta})$ denotes the final frequency of node $j$ after applying perturbation $\boldsymbol{\delta}$. Where we use the same threshold as in prior work. Cells with $\text{LBS}_{i,m,n} = 1$ are fully stable and cells with $\text{LBS}_{i,m,n} = 0$ are fully unstable. Stacking the $20 \times 20$ values yields $\mathbf{L}^{(i)} \in [0, 1]^{20 \times 20}$, the per-node heatmap label. While the individual error of $\text{LBS}_{i,m,n}$ is quite large, the overall landscapes that emerge are quite robust, see Figure 2.

Let $\mathcal{N}_i = \sum_{m,n} q_{i,m,n}$ be the number of trajectories for node $i$ and define $w_{i,m,n} = q_{i,m,n}/\mathcal{N}_i$. By construction,

$$\text{SNBS}_i(\mathcal{G}) = \sum_{m,n} w_{i,m,n} \text{LBS}_{i,m,n}, \tag{6}$$

for $\mathcal{N}_i = 10,000$ in our dataset. This makes the landscape

*Table 1.* Dataset statistics. SNBS is mean over all nodes. *Syn* denotes synthetic topologies and *Real* denotes real-world topologies.

| Type | Name | #Nodes | #Edges | #Graphs | $\overline{\text{SNBS}}$ |
|---|---|---|---|---|---|
| Syn | dataset20 | 20 | 538,188 | 10,000 | 0.8374 |
| | dataset100 | 100 | 2,857,882 | 10,000 | 0.8735 |
| Real | Germany | 438 | 1,324 | 1 | 0.9045 |
| | France | 146 | 446 | 1 | 0.8757 |
| | GB | 120 | 165 | 1 | 0.8368 |
| | Spain | 98 | 350 | 1 | 0.9288 |

strictly richer than the classical scalar while remaining backward compatible. Figure 2 contrasts raw point clouds with their corresponding downsampled heatmaps. As shown in Theorem A.1, reconstructing such a heatmap keeps the previous SNBS prediction error under control.

## 2.2. Properties of the dataset

The dataset consists of two ensembles: dataset20 and dataset100. Each sample is a triple $(\mathcal{G}, i, \mathbf{L}^{(i)})$, where $\mathcal{G} = (\mathcal{V}, \mathcal{E}, \mathbf{X})$ is an undirected graph, $i \in \mathcal{V}$ is the designated node, and $\mathbf{L}^{(i)} \in [0, 1]^{20 \times 20}$ is its basin-landscape heatmap label. $X_i = P_i \in \{+1, -1\}$ denotes the node feature, indicating power injection (see Equation (4)). Each ensemble is divided into training, validation, and test sets in a 70:15:15 ratio. Basic information about the datasets is summarized in Table 1. The histograms for the downstream task SNBS are shown in Figure 6 in Appendix A.1. Notably, for dataset100, there are more nodes that remain stable throughout. This difference in distribution poses a challenge for the out-of-distribution generalization task.

Furthermore, this dataset may help address the lack of benchmarks featuring extensive long-range dependencies, as the studied problem of dynamic stability inherently involves non-local behavior, and deeper GNNs demonstrate superior performance (Nauck et al., 2022a; 2023; 2024c;b; Raum et al., 2025). Combined with the task of image prediction, the dataset offers a challenging benchmark for developing GNN architectures capable of handling long-range dependencies. This is particularly relevant for applications such as power grids, where long-range interactions play a critical role (Ringsquandl et al., 2021).

## 3. Architecture: Encoder-decoder design and optimization

We formulate basin landscape prediction as a supervised mapping:

$$f_\theta : (\mathcal{G}, i) \longrightarrow \mathbf{L}^{(i)} \in [0, 1]^{20 \times 20}, \tag{7}$$

where each graph $\mathcal{G} = (\mathcal{V}, \mathcal{E}, \mathbf{X})$ carries one binary node feature $X_i = P_i \in \{-1, 1\}$ and $\mathbf{L}^{(i)}$ is the heatmap of a

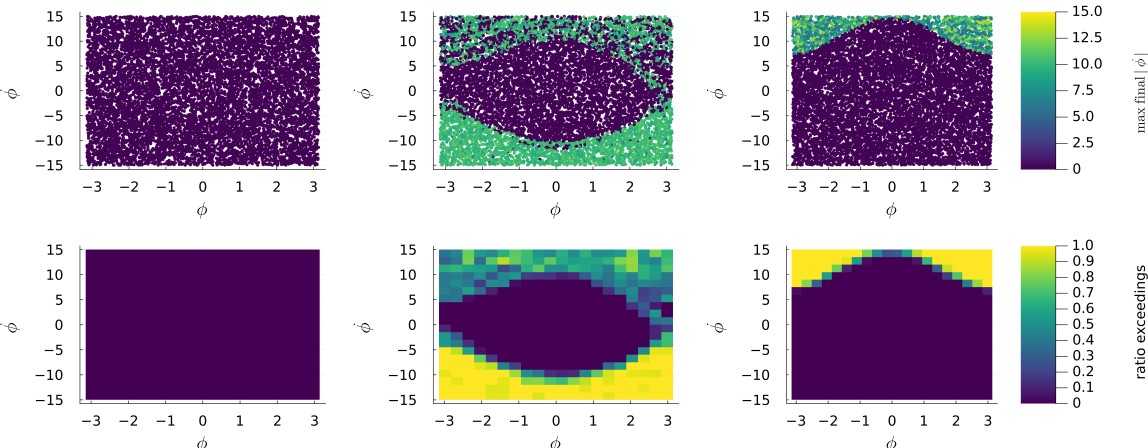

*Figure 2.* Examples of basin landscapes, original MC samples (top) and derived Basin Landscape with 20x20 grid cells (bottom).

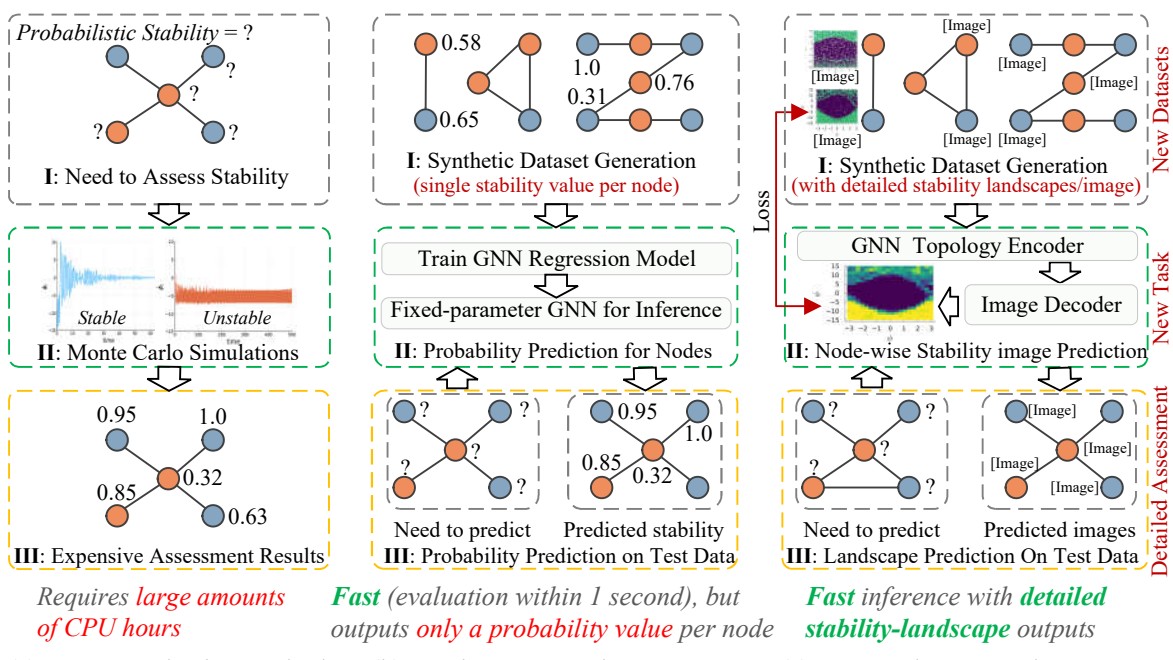

(a) Non-ML Physics Methods     (b) Previous ML Task & Dataset     (c) Proposed New Task & Dataset

*Figure 3.* Comparison of stability-assessment workflows. (a) Traditional physics-based methods rely on costly Monte Carlo simulations to evaluate single-node basin stability, requiring large amounts of CPU hours. (b) Prior ML approaches (Nauck et al., 2022b;a; 2023; Zhu et al., 2026) replace simulation with a trained GNN that predicts a probabilistic stability value per node in seconds, but sacrifices spatial detail. (c) In this work, we introduce a new dataset of full basin-landscape heatmaps and formulate a novel ML task: a GNN topology encoder plus image decoder jointly predict detailed, node-wise stability landscapes in one fast, end-to-end pass.

designated node. Figure 3 (c) illustrates our architecture, which places the model within the broader workflow of stability assessment.

## 3.1. Topology encoder and image decoder

**Topology encoder for embedding graph structural information.** The encoder $g_\phi$ is a message–passing graph neural network that converts the irregular topology into latent node embeddings $\mathbf{Z} = g_\phi(\mathcal{G}) \in \mathbb{R}^{|\mathcal{V}| \times d}$. Previous studies (Raum et al., 2025; Nauck et al., 2022a; 2023; 2024c;b) suggest that SNBS is not purely a local property but relies on long-range dependencies. The importance of such dependencies has also been independently observed in other power grid applications (Ringsquandl et al., 2021). Given this evidence, GNNs with long-range capabilities are well-suited to serve as the encoder $g_\phi$. In this study, we utilize Topology Adaptive Graph convolution (TAG) (Du et al., 2017) and Dirac–Bianconi GNN (DBGNN) (Nauck et al., 2024b) convolution. TAG uses learnable polynomial filters to aggregate information from up to $K = 3$ hops per layer. DBGNN incorporates multiple micro-propagation steps within each layer, allowing the model to efficiently capture multi-hop information in a single layer.

**Decoder for reconstructing basin landscape.** For the chosen node we extract its embedding $z_i \in \mathbb{R}^d$ and pass it to a lightweight MLP $h_\psi : \mathbb{R}^d \to \mathbb{R}^{400}$ or a convolutional neural network (CNN) decoder. The output is reshaped into a single-channel $20 \times 20$ image. Although advanced generative decoders, such as diffusion models and variational autoencoders, present promising avenues for future exploration, in this initial investigation we employ less complex per-node MLPs and CNNs to maintain model simplicity and support clear evaluation on our newly introduced task and dataset.

## 3.2. Learning objective

Ground-truth heatmaps contain continuous stability probabilities, and therefore training employs the mean-squared error (MSE):

$$\mathcal{L} = \left\| h_\psi(z_i) - \mathbf{L}^{(i)} \right\|_2^2, \quad \text{where } z_i = \left( g_\phi(\mathcal{G}) \right)_i. \quad (8)$$

Since SNBS equals the weighted average of the heatmap entries, this loss upper-bounds the downstream SNBS error, which is detailed in Theorem A.1 including its proof. The theorem guarantees that minimizing the weighted pixel-level MSE already keeps the SNBS error under control.

## 3.3. Contingency screening

The generated heatmaps enable a more detailed analysis for contingency screening, a key task for grid operators. By leveraging the stability landscape, it becomes straightforward to identify regions where minimal perturbations are most likely to destabilize the grid. These regions can then be prioritized by operators for further in-depth analysis to understand the underlying causes of system vulnerability—an approach that would be challenging to implement directly.

A common strategy involves analyzing the geometric features of the landscape space from which the system can recover to normal operation. This is a central topic in engineering and mechanics (Ma et al., 2019), with ongoing research in the area (Zhang & Strogatz, 2021). A typical focus is on determining the size of the smallest disturbance capable of destabilizing the system (Halekotte & Feudel, 2020). The probabilistic analogue—namely, the radius at which the probability of desynchronization exceeds a certain threshold—is studied as the linear size of the basin in (Delabays et al., 2017). The radial projection of our stability landscape is exactly what they estimate directly numerically. This could be an immediate downstream task using the predicted landscapes. From a power grid perspective, a further refinement is more interesting: Finding not just the radius, but also the precise direction of perturbation at which we first see a high likelihood of failure. In the power grid operator setting this provides a type of contingency screening: As not all possible contingencies can be studied, the basin landscape allows identifying a set of minimal perturbation regions that have a high likelihood of destabilizing the grid. These can then be singled out for further in-depth analysis, to understand why the system is prone to failure here.

To identify the most critical perturbations, we use the following procedure: For all pixels in the grid ($20 \times 20 \times N_{\text{nodes}}$), we select those with an exceeding ratio greater than 0.7. Among these, we identify the 20 most critical cells by lexicographic sorting—first by distance to the center, then by exceeding ratio.

## 4. Predictive performance

The predictive performance of the model is assessed in four setups. First, we evaluate the model's ability to predict heatmaps both qualitatively and quantitatively. Second, we evaluate a downstream scalar task by recovering SNBS from the heatmaps through Equation (3). Third, zero-shot transfer to four real grid topologies, namely Germany, France, Great Britain, and Spain are analyzed. Unless noted otherwise, zero-shot uses the model trained on *dataset100*. Fourth, we introduce a novel task to identify critical contingencies. A brief hyperparameter study was conducted to tune the model and training parameters. The resulting model properties are provided in Appendix A.5.

The DBGNN uses two layers, each with 10 internal propagation steps (alternating between node-to-line and line-to-node

*Table 2.* Performance on predicting the heatmaps of the landscapes measured by $SSIM$ in %.

|  | Model | In-Distribution | | Out-of-Distribution |
|---|---|---|---|---|
|  |  | tr20ev20 | tr100ev100 | tr20ev100 |
| MLP | TAG | $81.37_{\pm1.58}$ | $81.41_{\pm0.23}$ | $71.11_{\pm1.47}$ |
| MLP | DBGNN | $82.63_{\pm1.75}$ | $83.62_{\pm0.90}$ | $75.49_{\pm2.44}$ |
| CNN | TAG | $84.31_{\pm0.22}$ | $83.54_{\pm0.10}$ | $74.36_{\pm0.30}$ |
| CNN | DBGNN | $\mathbf{86.69}_{\pm1.91}$ | $\mathbf{84.90}_{\pm0.33}$ | $\mathbf{81.03}_{\pm1.59}$ |

*Table 3.* Performance on the downstream task of predicting SNBS based on heatmap predictions measured by $R^2$ in %. Baselines are from (Nauck et al., 2023; 2024b).

|  | Model | In-Distribution | | Out-of-Distribution |
|---|---|---|---|---|
|  |  | tr20ev20 | tr100ev100 | tr20ev100 |
|  | ArmaNet | $82.22_{\pm0.12}$ | $88.35_{\pm0.12}$ | $67.12_{\pm0.80}$ |
|  | GCNNet | $70.74_{\pm0.15}$ | $75.19_{\pm0.14}$ | $58.24_{\pm0.47}$ |
|  | TAGNet | $82.50_{\pm0.36}$ | $88.32_{\pm0.10}$ | $66.32_{\pm0.74}$ |
|  | DBGNN | $\mathbf{85.68}_{\pm0.10}$ | $\mathbf{90.08}_{\pm0.02}$ | $\mathbf{73.73}_{\pm0.07}$ |
|  | TAG | $82.61_{\pm0.46}$ | $86.60_{\pm0.50}$ | $60.48_{\pm1.39}$ |
|  | DBGNN | $83.99_{\pm0.04}$ | $89.16_{\pm0.07}$ | $70.66_{\pm1.83}$ |
| CNN | TAG | $80.86_{\pm0.84}$ | $86.06_{\pm0.16}$ | $56.25_{\pm0.75}$ |
| CNN | DBGNN | $83.96_{\pm0.09}$ | $88.75_{\pm0.10}$ | $70.52_{\pm1.64}$ |

updates), effectively capturing information from nodes up to 10 edges away. For TAG, the parameter ( $K$=3 ) is selected with 5 layers, allowing the TAG model to consider nodes within 15 edge steps. Further information on the models and the training settings is provided in Appendix A.5.

### 4.1. Evaluation: heatmap comparison

The qualitative performance of the ML model is assessed by comparing the predicted heatmaps to the ground truth. Example heatmaps from the TAG-MLP model, trained and evaluated on dataset20, are shown in Figure 4. Additional results for TAG-MLP trained and evaluated on dataset100, as well as for the out-of-distribution (OOD) task, trained on dataset20 and evaluated on dataset100, are provided in the appendix (Appendix A.6). Furthermore, we demonstrate the models' generalization capabilities on the real-world grid topologies of France, Germany, Great Britain, and Spain, as discussed in Section 4.3.

Overall, the visualizations demonstrate that the general shapes of the landscapes are often predicted accurately. However, there are instances where the predicted SNBS deviates significantly from the true labels, highlighting areas for improvement. These discrepancies can occur even when the qualitative comparison shows that the stable region's structure is correctly predicted, but the ratio of exceedances is not accurately captured. Thus, even if the predicted SNBS differs from the true value, the prediction may still be valuable as it correctly identifies the stable region.

The quantitative performance, summarized in Table 2 using the structural similarity index measure (SSIM), demonstrates that the heatmaps can be predicted with an SSIM of up to 86.7%. Importantly, the approach also performs well in out-of-distribution generalization, albeit with noticeably lower performance. The best performance is achieved using a CNN as image decoder. Additional metrics are provided in Appendix A.7.

### 4.2. Evaluation on the downstream task of recovering SNBS from predicted heatmaps

As a second benchmark, we evaluate the downstream scalar task obtained by recovering SNBS from the predicted heatmaps. Similar to (Nauck et al., 2023), we report the mean value of the best 3 seeds out of 5 initializations and the corresponding standard deviation. The performance of the downstream task is presented in Table 3. The overall performance is comparable to the results achieved when directly predicting SNBS, bypassing the intermediate step of predicting the landscapes. The slightly lower performance observed may stem from the increased complexity of accurately predicting the landscapes. Examining exemplary basin landscapes in Figures 4 and 9, we often observe symmetric structures with comparable probabilities. By symmetric structures, we refer to regions where stability is present either at low $\phi$ or high $\phi$, with only a small region in between where instabilities occur. When predicting only the probability, it suffices to approximate the size of this unstable region. However, predicting the landscapes requires not only estimating the size, but also accurately locating the unstable region. In contrast, for probability prediction, the specific location of the stable regions (whether at low or high $\phi$) is irrelevant. This added requirement of spatial precision in landscape prediction likely contributes to the observed reduction in overall performance. To conclude, the performance reduction is minimal, and the results serve as a proof of concept, highlighting the potential of using ML to predict landscapes.

### 4.3. Generalization to real-world topologies

To further evaluate generalization, we assess model performance on the real-world grid topologies of France, Germany, Great Britain, and Spain. As shown in Table 4, the ML models generally adapt well to these realistic networks. TAG

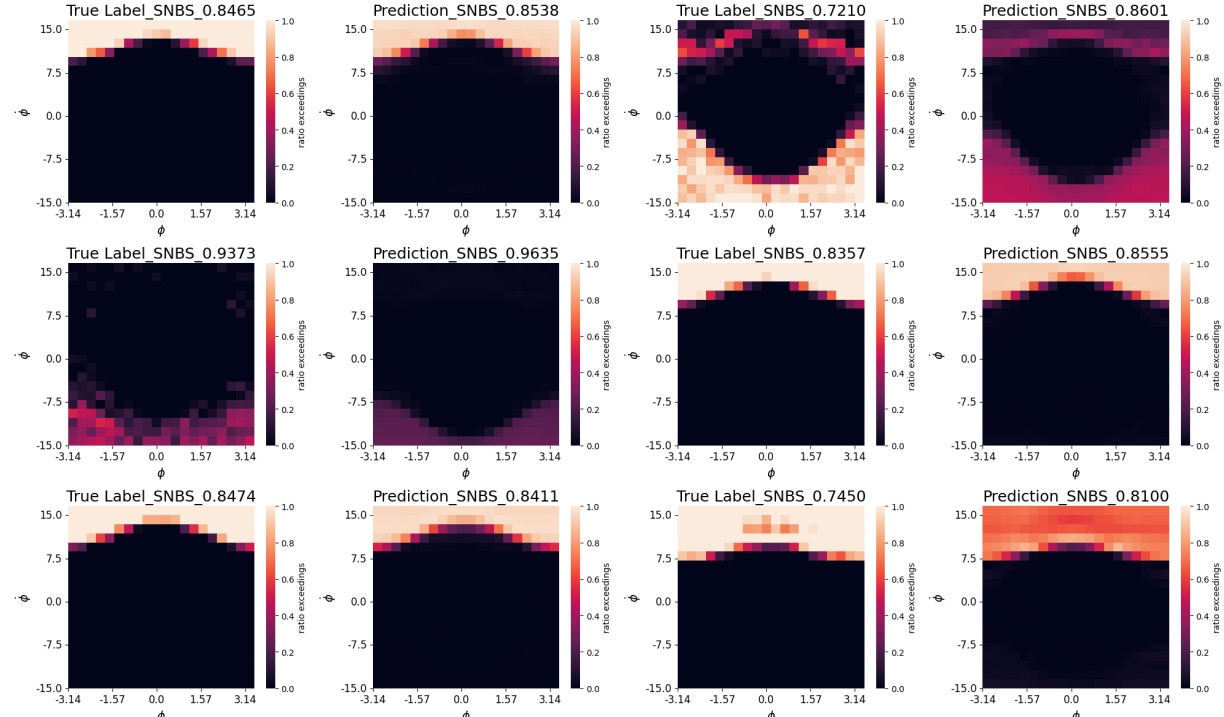

*Figure 4.* Comparison of true labels and predicted heatmaps for the TAG-MLP model trained and evaluated on dataset20.

*Table 4.* Out-of-distribution performance on predicting the heatmaps of real topologies measured by $SSIM$ in %.

| | Model | Germany | France | GB | Spain |
|---|---|---|---|---|---|
| MLP | TAG | $80.79_{\pm0.24}$ | $76.76_{\pm0.69}$ | $81.84_{\pm0.30}$ | $75.93_{\pm1.23}$ |
| | DBGNN | $54.77_{\pm4.05}$ | $53.41_{\pm2.29}$ | $54.22_{\pm1.80}$ | $60.23_{\pm6.76}$ |
| CNN | TAG | $\mathbf{82.87}_{\pm0.26}$ | $\mathbf{79.26}_{\pm0.45}$ | $\mathbf{85.38}_{\pm0.83}$ | $\mathbf{79.45}_{\pm0.09}$ |
| | DBGNN | $53.10_{\pm5.10}$ | $50.46_{\pm2.86}$ | $45.38_{\pm0.23}$ | $59.29_{\pm4.02}$ |

models, in particular, exhibits robust generalization to grids with hundreds of nodes, despite being trained solely on small synthetic graphs, suggesting strong scalability. The performances on the down-stream task are provided in Table 13.

### 4.4. Identification of critical contingencies

Predicting heatmaps enables fine-grained contingency screening by identifying the most critical perturbations. We use the predicted heatmaps as the basis for a novel downstream task: ranking and detecting the most critical contingencies. Figure 5 highlights the most critical cells using lime circles. The screening pinpoints the smallest yet most critical perturbations, demonstrating the potential of utilizing ML to preselect contingencies for further analysis.

Quantitative assessment is challenging because we aim to predict individual cells (pixels) rather than broader regions. For instance, predicting a neighboring cell may be accept-

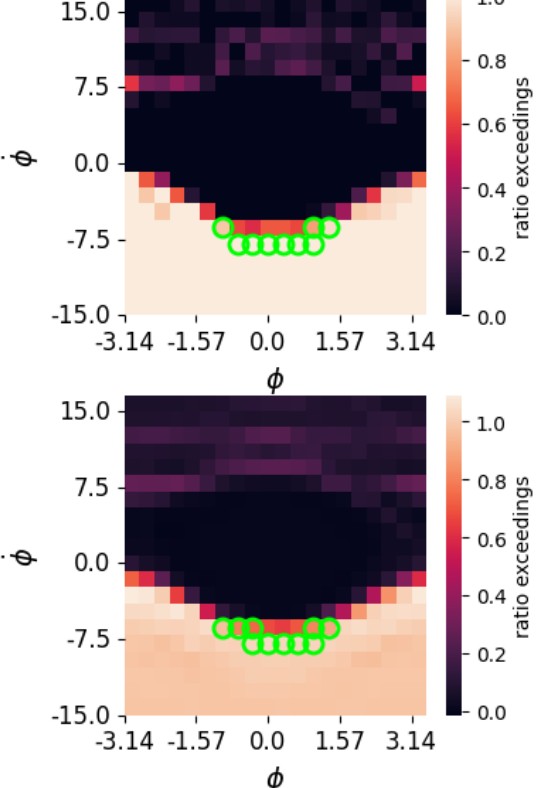

*Figure 5.* Visualization of critical contingencies (true labels on the top and predictions on the bottom) of a node from dataset20. The predictions are from TAG-MLP.

*Table 5.* Performance on the downstream task of identifying critical contingencies, measured by intersection over union between predicted and true critical cells.

| | Model | In-Distribution | | Out-of-Distribution |
|---|---|---|---|---|
| | | tr20ev20 | tr100ev100 | tr20ev100 |
| MLP | TAG | $0.58_{\pm 0.01}$ | $0.41_{\pm 0.01}$ | $0.32_{\pm 0.02}$ |
| | DBGNN | $0.58_{\pm 0.02}$ | $0.50_{\pm 0.00}$ | $0.38_{\pm 0.03}$ |
| CNN | TAG | $0.58_{\pm 0.01}$ | $0.47_{\pm 0.02}$ | $0.36_{\pm 0.01}$ |
| | DBGNN | $\mathbf{0.60}_{\pm 0.03}$ | $\mathbf{0.52}_{\pm 0.00}$ | $\mathbf{0.40}_{\pm 0.04}$ |

able if it has similar stability, but it can be problematic if the neighboring cell is actually stable. Therefore, we use two metrics: i) a strict evaluation procedure that only considers a prediction correct if the exact same cell is identified, ii) a relaxed evaluation metric for which we compute an accuracy by evaluating whether the 20 most critical cells according to the ground truth labels fall within the 30 most critical cells according to the model predictions. For the latter, the performances are provided in Appendix A.9.

For the strict metric, predictive in-distribution performance is reported in Table 5, using the intersection over union metric, which quantifies the overlap between the 20 most critical cells in the predicted and true heatmaps. The out-of-distribution performance is reported in Table 14.

## 5. Limitations

Our work is subject to several important limitations that should be considered when interpreting the results. First, we focus on homogeneous Kuramoto oscillator networks with uniform coupling and parameters. While these systems are of theoretical interest and provide a foundation for understanding synchronization dynamics, they do not directly represent real-world power grids or other practical applications. The underlying ensembles are inspired by power grid topologies but incorporate simplifying assumptions that limit direct applicability. Nevertheless, the synchronization problem remains fundamentally relevant, and our landscape-based analysis has broader significance for the physics community.

Second, our pipeline is evaluated on a single oscillator model with homogeneous parameters. Extending the analysis to other dynamical systems and heterogeneous network properties would strengthen the generalizability of our approach. However, since all oscillators share a common underlying structure (topology, parameters, and line properties), scaling to additional dynamical parameters should pose no fundamental challenge to our machine learning framework. This is evidenced by recent work that success-

fully incorporates heterogeneous node and edge features with different dynamical actors while predicting stability properties in more realistic power grid models (Nauck et al., 2024a).

Third, some of the real power grids analyzed in this work feature small numbers of nodes, which limits the statistical power for evaluating model generalization through direct benchmarking across different models. Nevertheless, demonstrating successful generalization to such constrained systems provides compelling evidence that machine learning models can learn underlying physical principles across different networks. Additionally, more sophisticated machine learning decoding architectures could further enhance predictive performance beyond the methods presented here.

Despite these limitations, we believe this work represents an important contribution to the field. The primary bottleneck of our approach is data scarcity—to our knowledge, no publicly available datasets currently exist for validating this pipeline. We hope this work serves as a proof of concept, encouraging the research community to invest in generating large-scale datasets across related domains.

## 6. Conclusion

We introduce graph-to-field prediction of per-node stability landscapes from topology, replacing a single SNBS score with a field that localizes where failures arise. We release two large-scale datasets with landscape labels at 20 and 100 nodes that support in-distribution testing and size-shift evaluation. Our models with GNN encoders match the fidelity of Monte Carlo while reducing evaluation from thousands of CPU hours across the datasets to seconds per graph. They generalize without retraining to four real power grid topologies. These pieces could establish a practical benchmark and a template for learning stability landscapes on graphs, and they invite landscape-level prediction across other domains. We expect the framework to transfer to other dynamical systems (biochemical, neuroscience, and infrastructure) given sufficient high-resolution training data.

## Software and data

All code for dataset generation, model training, and analysis is made publicly available on GitHub (Nauck et al., 2025d) and Zenodo (Nauck et al., 2025a). Ready-to-use datasets for machine learning training will be hosted on HuggingFace (Nauck et al., 2025c). The repository includes tools for training, evaluation, hyperparameter optimization, and visualization, with instructions for environment setup and experiment replication. The raw simulation data, from which landscapes at arbitrary resolution can be reconstructed, is available at (Nauck et al., 2025b).

## Impact Statement

This paper presents work whose goal is to advance the field of Machine Learning. There are many potential societal consequences of our work, none which we feel must be specifically highlighted here.

## Acknowledgements

All authors gratefully acknowledge Land Brandenburg for supporting this project by providing resources on the high-performance computer system at the Potsdam Institute for Climate Impact Research. The work was in part supported by BMBF Grant 03SF0766, and BMWK grant 03EI1092A. Christian Nauck would like to thank Professor Jörg Raisch for supervising his PhD. We especially want to thank the reviewers for their valuable feedback and ideas, which significantly improved the quality of this paper.

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

## A. Technical Appendices and Supplementary Material

The appendix is organized into four sections. The first section provides additional information on the datasets. The second section presents the theoretical proof for the theorem concerning the relationship between the heatmaps and SNBS. Next, details regarding the models and training procedures are described. Finally, the appendix concludes with supplementary results.

### A.1. Further Properties of the Dataset

Figure 6 shows the histograms for the downstream task target SNBS. In comparison to dataset20, the ensemble with graphs of size 100 contains more nodes that remain stable under all perturbations. This difference in distribution poses a challenge for machine learning models to generalize across ensembles of varying sizes. The out-of-distribution challenge is especially challenging for the real-world topologies, since they show a different SNBS distribution.

**Justification of used threshold for categorizing trajectories as stable** To categorize trajectories as stable or unstable a threshold of $0.1$ is used for the maximum final frequency deviation. There is essentially no sensitivity to this threshold. Asymptotically the system either synchronizes or runs towards a limit cycle with phase velocity around 10/s. Earlier studies of re-synchronization explored sophisticated heuristics to robustly identify the cases where we have re-synchronization, but experience shows that the simple threshold 0.1 is sufficient and achieves identical probabilistic results in the setting explored here. We present a histogram of the maximum final frequency deviation and indicated the 0.1 threshold in Figure 7. Almost no configurations lie just above this cutoff. Because the same threshold was also used as an early-stopping criterion for synchronization, values on the synchronized side of the boundary cannot be resolved in finer detail. This is, however, consistent with current state-of-the-art practice in oscillator-network studies.

### A.2. Robustness of simulations on the cell level

Each node receives a different set of random perturbations, resulting in varying perturbation distributions across cells (including variation in the number of perturbations per cell). While our dataset contains several different perturbation distributions, we did not systematically study distinct sets for identical nodes. The uncertainties per cell are substantial due to the relatively sparse sampling—on average, only 25 perturbations per cell for 20×20 image resolution. As an illustration, in a Bernoulli setting with $p = 0.5$ and $n = 25$, the empirical proportion has standard deviation $\sqrt{p(1-p)/n} = 0.1$. In particular, cells with p close to

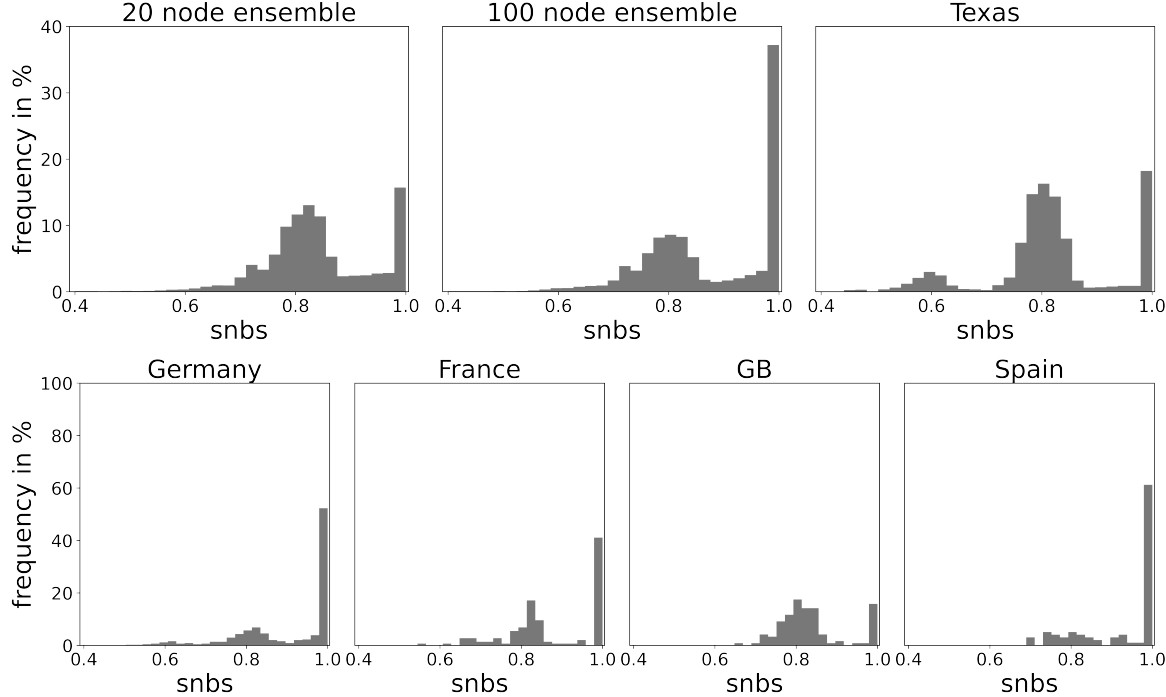

*Figure 6.* Histograms of the downstream task SNBS. The top panel corresponds to the synthetic topologies: dataset20, dataset100 and Texas. The bottom represents the real-world topologies. Figures from (Nauck et al., 2024c)

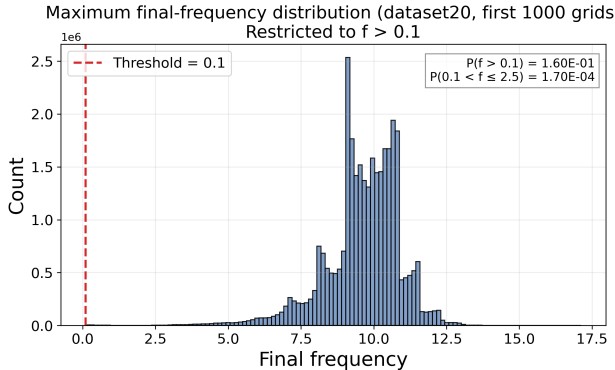

*Figure 7.* Distribution of final frequencies for the first 100 grids of dataset20. The linear histogram is limited to samples with $f > 0.1$. The legend probabilities show the fractions of samples with $f > 0.1$ and $0.1 < f < 2.5$, confirming that these results do not affect the reported statistics. The red dashed vertical line marks the synchronization threshold at $f = 0.1$.

0 or 1 will have almost no dependence on the within-cell distribution. The location of the boundary between stable and unstable regions will not change meaningfully. Overall, when predicting across a large set of cells spanning multiple nodes and grids, the inherent sparsity can be viewed as noise, which the machine learning models appear to handle effectively.

### A.3. Details on dataset generation and numerical simulations

The synthetic topologies are generated using the random-growth algorithm of (Schultz et al., 2014), which is designed to reproduce key topological characteristics of power grids, including a realistic degree distribution. We use the same parameter setting as in (Nitzbon et al., 2017; Nauck et al., 2022b): initial number of nodes $n_0 = 1$, $p = \frac{1}{5}$, $q = \frac{3}{10}$, $r = \frac{1}{3}$, and $s = \frac{1}{10}$. Here, $p$ and $q$ determine the probabilities of adding new transmission lines, $s$ is the probability of splitting an existing line, and $r$ controls the creation of redundant paths.

To solve the differential equations, the Julia package DifferentialEquations.jl (Rackauckas & Nie, 2017) and the solver Tsit5 (Tsitouras, 2011) is used. We run the simulations for 500 seconds, with early stopping in case of convergence. The synchronous static state is computed by solving the nonlinear nodal balance equations, where node phases are mapped to complex phasors and network flows are derived

from the Laplacian. The root finder nlsolve from the NL-Solve.jl package starts from a pseudoinverse-based initial guess.

## A.4. Bounding SNBS error by pixel MSE

**Theorem A.1** (Pixel MSE upper-bounds SNBS error). *Fix a node $i$. For each grid cell $(m, n)$, let $q_{i,m,n} \geq 0$ be the number of trajectories in that cell and set $\mathcal{N}_i = \sum_{m,n} q_{i,m,n}$ and $w_{i,m,n} = q_{i,m,n}/\mathcal{N}_i$. Let $LBS_{i,m,n}, \widehat{LBS}_{i,m,n} \in [0, 1]$ be the true and predicted stability probabilities. Define*

$$\text{SNBS}_i = \sum_{m,n} w_{i,m,n} \, LBS_{i,m,n}, \tag{9}$$

$$\widehat{\text{SNBS}}_i = \sum_{m,n} w_{i,m,n} \, \widehat{LBS}_{i,m,n}.$$

*Then*

$$\left(\widehat{\text{SNBS}}_i - \text{SNBS}_i\right)^2 \leq$$
$$\sum_{m,n} w_{i,m,n} \left(\widehat{LBS}_{i,m,n} - LBS_{i,m,n}\right)^2. \tag{10}$$

*The right-hand side is exactly the weighted pixel-wise MSE employed as the loss in Equation* (8) *when sample counts per cell are unequal.*

Theorem A.1 guarantees that minimizing the weighted pixel-level MSE already keeps the SNBS error under control. Consequently we do not need to append a second loss term such as $(\widehat{\text{SNBS}}_i - \text{SNBS}_i)^2$ to the training objective. Omitting this extra term also removes the weighting factor that would otherwise be required to balance two losses, so optimization remains a single-objective problem with one fewer hyper-parameter to tune. Moreover, the inequality provides a quantitative error guarantee. Even if the model makes large mistakes on a small subset of pixels, the global SNBS error cannot exceed the reported weighted-MSE. This gives an interpretable upper bound: as long as the overall heatmap MSE is small, the classical stability score is automatically protected.

The following presents the proof of Theorem A.1.

*Proof.* Let $\varepsilon_{i,m,n} = \widehat{LBS}_{i,m,n} - LBS_{i,m,n}$ and recall that the weights $w_{i,m,n} \geq 0$ satisfy $\sum_{m,n} w_{i,m,n} = 1$. By the

definitions of (weighted) SNBS we have:

$$\left(\widehat{\text{SNBS}}_i - \text{SNBS}_i\right)^2$$
$$= \left(\sum_{m,n} w_{i,m,n} \, \widehat{LBS}_{i,m,n} - \sum_{m,n} w_{i,m,n} \, \text{LBS}_{i,m,n}\right)^2$$
$$= \left(\sum_{m,n} w_{i,m,n} \, \epsilon_{i,m,n}\right)^2$$
$$\leq \sum_{m,n} w_{i,m,n} \, \epsilon_{i,m,n}^2 \quad \text{(by Jensen's inequality)}$$
$$= \sum_{m,n} w_{i,m,n} \left(\widehat{LBS}_{i,m,n} - \text{LBS}_{i,m,n}\right)^2$$
$$= \sum_{m=1}^{20} \sum_{n=1}^{20} w_{i,m,n} \left(\widehat{LBS}_{i,m,n} - \text{LBS}_{i,m,n}\right)^2, \tag{11}$$

which completes the proof. $\qquad\square$

## A.5. Model and training properties

The DBGNN uses two layers, each with 10 internal propagation steps (alternating between node-to-line and line-to-node updates), effectively capturing information from nodes up to 10 edges away. For TAG, the parameter ($K$=3) is selected with 5 layers, allowing the TAG model to consider nodes within 15 edge steps. Details of the final model architecture are summarized in Table 6, while the training properties are listed in Table 7. The CNN decoder replaces the MLP head with a three-layer transposed-convolutional decoder (channels $64 \to 32 \to 16$, single output channel, no dropout), while the GNN encoders remain unchanged. Hyperparameter optimization was performed using Optuna (Akiba et al., 2019). Training was conducted on a local HPC equipped with an Nvidia A100 GPU (80GB memory). For DBGNN-MLP, training on dataset20 with 5 consecutive seeds required approximately 21 hours and 35 minutes, while training on dataset100 took about 6 days. For TAG-MLP, training on dataset20 took less than 12 hours, and on dataset100 less than 2 days.

Table 8 presents the inference times, demonstrating the huge advantage over running Monte Carlo simulations requiring several hours per grid.

## A.6. Additional Results

Example heatmaps for TAG-MLP trained and evaluated on dataset100 are shown in Figure 8, while heatmaps for TAG-MLP trained on dataset20 and evaluated on dataset100 are presented in Figure 9. Consistent with the observations from Figure 4, the predictions capture most of the underlying structure accurately.

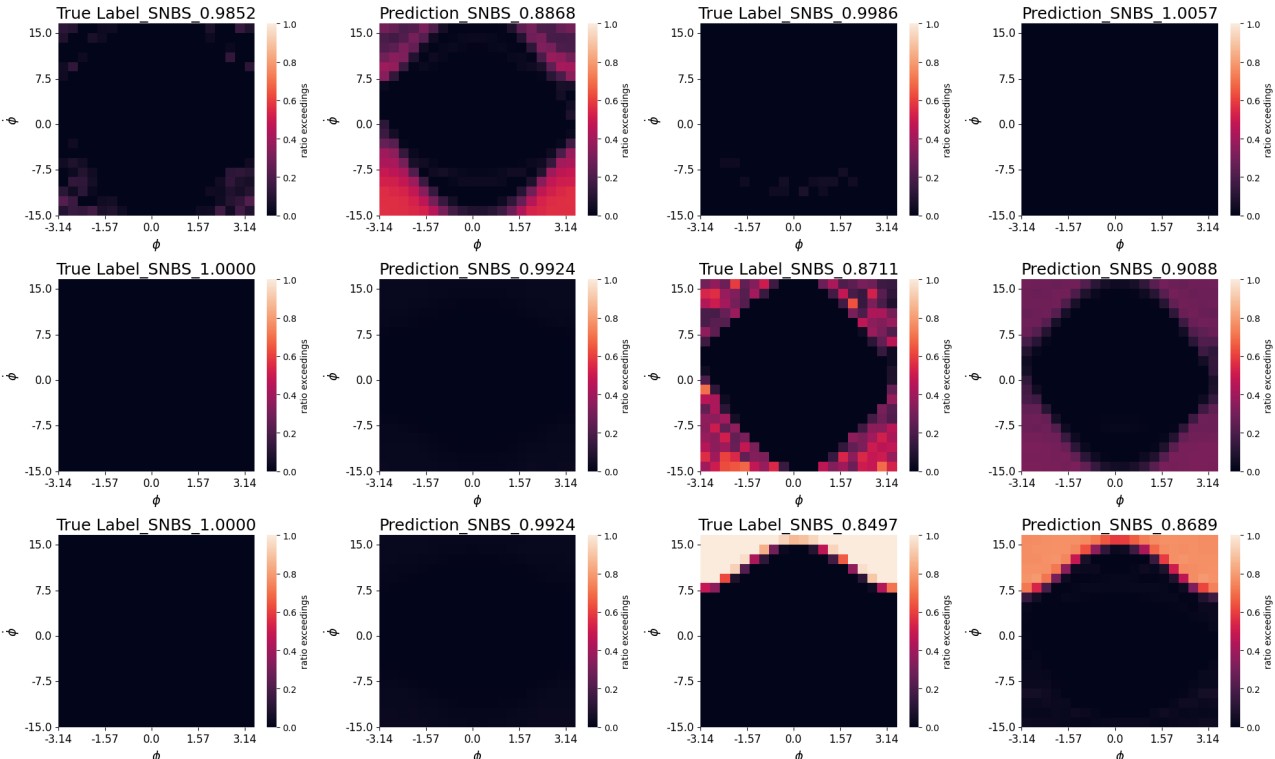

*Figure 8.* Comparison of true labels and predicted heatmaps for the TAG-MLP model trained and evaluated on dataset100.

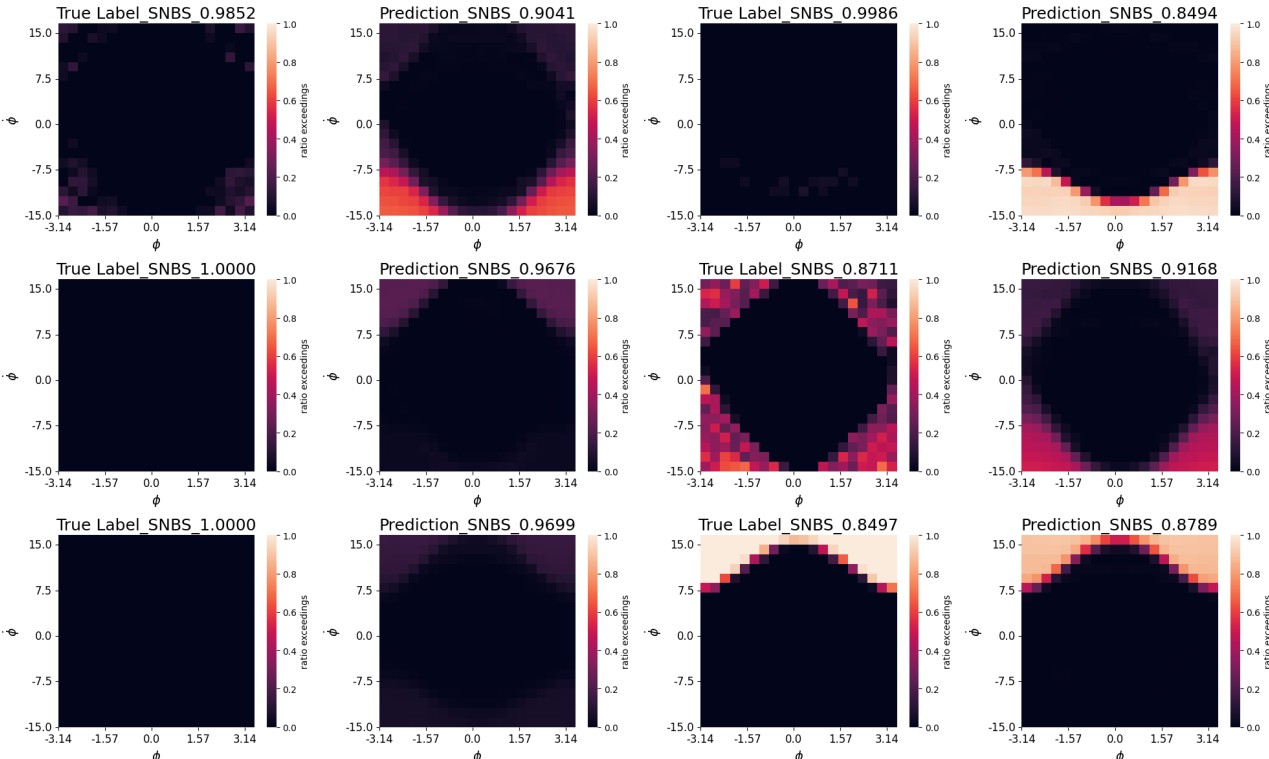

*Figure 9.* Comparison of true labels and predicted heatmaps for the TAG-MLP model trained on dataset20 and evaluated on dataset100.

*Table 6.* Summary of model properties.

| Property | TAG-MLP | DBGNN-MLP |
|---|---|---|
| Number of GNN layers | 5 | 2 |
| Hidden dimension of GNN | 1751 | 512 |
| Output dimension of GNN | 671 | 512 |
| Hidden dimension of MLP | 654 | 512 |
| Output dimension of MLP | 512 | 512 |
| Dropout of GNN layer | $\approx 0.148$ | 0.04 |
| Dropout of MLP layer | 0.26 | 0.02 |

*Table 7.* Overview of training properties. MSE denotes mean squared error. The TAG–CNN and DBGNN–CNN decoders (trained on *dataset20* and *dataset100*) use the hyperparameters shown in the last column.

| Parameters | TAG-MLP (tr20) | TAG-MLP (tr100) | DBGNN-MLP (tr20) | DBGNN-MLP (tr100) | CNN decoders |
|---|---|---|---|---|---|
| Learning rate | $\approx$ 7.77e-05 | $\approx$ 6.77e-05 | $\approx$ 6.77e-06 | $\approx$ 6.77e-05 | $\approx$ 6.77e-05 |
| Number of epochs | 250 | 250 | 1000 | 1000 | 250 |
| Loss function | MSE | MSE | MSE | MSE | MSE |
| Training batch size | 32 | 32 | 32 | 32 | 32 |

*Table 8.* Inference times using the runtime for evaluating the full test sets (1,500 grids). This shows that 1,500 grids can be analyzed in under 2 seconds.

| Model | Runtime ds20 test split [s] | Runtime ds100 test split [s] |
|---|---|---|
| DBGNN-MLP | 0.364 | 1.61 |
| TAG-MLP | 0.288 | 0.795 |
| DBGNN-CNN | 0.39 | 1.923 |
| TAG-CNN | 0.298 | 1.004 |

## A.7. Additional results on quantitative comparison of heatmaps

In addition to Table 2, we also provide the performances measured by $R^2$ in Table 9 and Learned Perceptual Image Patch Similarity (LPIPS) Table 10.

In Tables 11 and 12, the out-of-distribution performances of the generated heatmaps are shown. The results confirm the overall observations from $SSIM$ from Table 4.

## A.8. Downstream task: Predicting SNBS

The performances on the down-stream task are provided in Table 13. The low downstream performance of DBGNN-MLP and the CNN decoders for SNBS prediction likely reflects the unique dynamical properties of the Great Britain grid (Nauck et al., 2024c), and underscores the complexity of the task.

## A.9. Downstream task: Identification of contingencies

Figure 5 visualized the most critical node in the first grid of dataset20. By also showing the second node, where critical cells are identified in Figure 10 all of the most 20 critical cells are visualized, since they only occurred at those two nodes.

Tables 15 and 16 report the less strict metric for which the proportion of cases in which all 20 most critical cells appear within the predictor's top 30. This approach softens the original strict requirement while remaining practically relevant, as it tests whether the model can identify the most critical cells within a reasonably constrained set.

## A.10. Investigating the impact of grid resolution

To evaluate the effect of grid resolution on the analysis, we test various cell densities beyond the standard $20 \times 20$ configuration. Figure 11 shows heatmaps generated with three different resolutions: 5, 10, 20, and 30 cells per axis. The $20 \times 20$ grid represents an good balance between capturing fine spatial structures and maintaining sufficient sample density within each cell for statistical reliability.

The performances in Tables 17 to 19 and Table 20 confirm that the approach is feasible with different grid sizes. The reached performance levels are similar.

*Table 9.* Performance on predicting the heatmaps of the landscapes measured by $R^2$ in %.

| Model | In-Distribution | | Out-of-Distribution |
|---|---|---|---|
| | tr20ev20 | tr100ev100 | tr20ev100 |
| TAG-MLP | $89.77_{\pm0.15}$ | $88.42_{\pm0.31}$ | $68.53_{\pm0.50}$ |
| DBGNN-MLP | $90.39_{\pm0.09}$ | $90.40_{\pm0.03}$ | $74.27_{\pm2.47}$ |
| TAG-CNN | $88.87_{\pm0.29}$ | $87.76_{\pm0.10}$ | $65.35_{\pm0.65}$ |
| DBGNN-CNN | $90.26_{\pm0.06}$ | $90.08_{\pm0.04}$ | $73.14_{\pm2.27}$ |

*Table 10.* Performance on predicting the heatmaps of the landscapes measured by $LPIPS$ in %.

| Model | In-Distribution | | Out-of-Distribution |
|---|---|---|---|
| | tr20ev20 | tr100ev100 | tr20ev100 |
| TAG-MLP | $0.16_{\pm0.00}$ | $0.15_{\pm0.00}$ | $0.22_{\pm0.00}$ |
| DBGNN-MLP | $0.16_{\pm0.02}$ | $0.14_{\pm0.00}$ | $0.20_{\pm0.03}$ |
| TAG-CNN | $0.14_{\pm0.00}$ | $0.14_{\pm0.00}$ | $0.20_{\pm0.00}$ |
| DBGNN-CNN | $0.12_{\pm0.01}$ | $0.13_{\pm0.00}$ | $0.15_{\pm0.01}$ |

Furthermore, we analyze the impact of the grid resolution on the down-stream task of identifying critical contingencies in Table 21. As expected, performance improves at lower resolution (fewer cells).

### A.11. Training curves of TAG-MLP and DBGNN-MLP

Figure 12 shows that TAG training is much smoother, reaching reasonable performance within a few steps, whereas DBGNN training is more difficult early on (including negative Image in the first epochs). Later, in-distribution performance improves rapidly, while generalization to real-world topologies saturates. This suggests that TAG is smoother overall, and smoother models are often associated with better generalization.

However, we do not necessarily believe the different out-of-distribution performance is a property of the layer itself, but maybe simply related to the specific hyperparameters in this study. Future work can test different hyperparameters to see what model properties show increased out-of-distribution generalization.

### A.12. Influence of size of training samples

Table 22 reports how performance (SSIM, %) varies with the number of training grids. Increasing the number of grids improves accuracy, while the method remains effective even with fewer training grids.

In the literature, prior work on basin-stability prediction (Fig. 4 in (Nauck et al., 2024c)) shows that GNN performance generally improves with larger training sets, while still remaining viable with fewer samples. That study also suggests that the key driver is the number of training nodes rather than the number of distinct grids. In this sense, our comparison between dataset20 and dataset100 provides an implicit test of training-set size effects: dataset100 contains roughly five times more training nodes. This difference is likely an important contributor to the performance gap between tr100ev100 and tr20ev20.

### A.13. Ablation study to show long-range dependencies

To demonstrate the advantage of models that enable considering long-range dependencies, we conduct ablation studies on the number of TAG layers for dataset20 and dataset 100. Table 23 demonstrates the advantage of using more than 5 convolutional layers.

Previous work already demonstrated the advantage of models capable of considering long-range dependencies ((Nauck et al., 2023; 2024b)), such as TAG and DBGNN outperforming GCNs.

*Table 11.* Out-of-distribution performance on predicting the heatmaps of real topologies measured by $R^2$ in %.

| Model | tr100evGermany | tr100evFrance | tr100evGB | tr100evSpain |
|---|---|---|---|---|
| TAG-MLP | 85.81 $\pm0.71$ | 81.01 $\pm2.11$ | 85.51 $\pm1.29$ | 66.69$\pm3.44$ |
| DBGNN-MLP | 50.00 $\pm5.58$ | 56.98$\pm4.42$ | 64.68$\pm3.13$ | 48.99$\pm16.04$ |
| TAG-CNN | 86.36$\pm0.51$ | 82.40$\pm0.50$ | 87.15$\pm0.11$ | 67.41$\pm1.40$ |
| DBGNN-CNN | 38.12$\pm4.91$ | 53.50$\pm3.77$ | 56.18$\pm5.67$ | 37.61$\pm8.83$ |

*Table 12.* Out-of-distribution performance on predicting the heatmaps of real topologies measured by $LPIPS$.

| Model | tr100evGermany | tr100evFrance | tr100evGB | tr100evSpain |
|---|---|---|---|---|
| TAG-MLP | 0.14 $\pm0.00$ | 0.18 $\pm0.00$ | 0.15 $\pm0.00$ | 0.17 $\pm0.01$ |
| DBGNN-MLP | 0.30 $\pm0.02$ | 0.31 $\pm0.02$ | 0.31 $\pm0.00$ | 0.27 $\pm0.04$ |
| TAG-CNN | 0.14 $\pm0.00$ | 0.16 $\pm0.00$ | 0.12 $\pm0.00$ | 0.16 $\pm0.00$ |
| DBGNN-CNN | 0.30$\pm0.03$ | 0.32$\pm0.02$ | 0.33$\pm0.01$ | 0.27$\pm0.03$ |

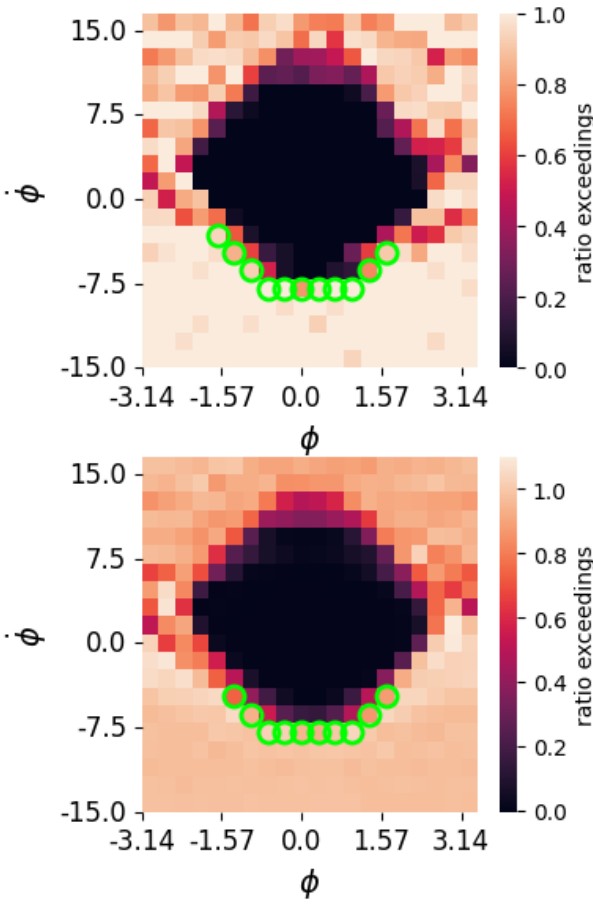

*Figure 10.* Visualization of critical contingencies for node 16 in the first grid of the test set from dataset20. True labels are shown on the top, and predictions are shown on the bottom.

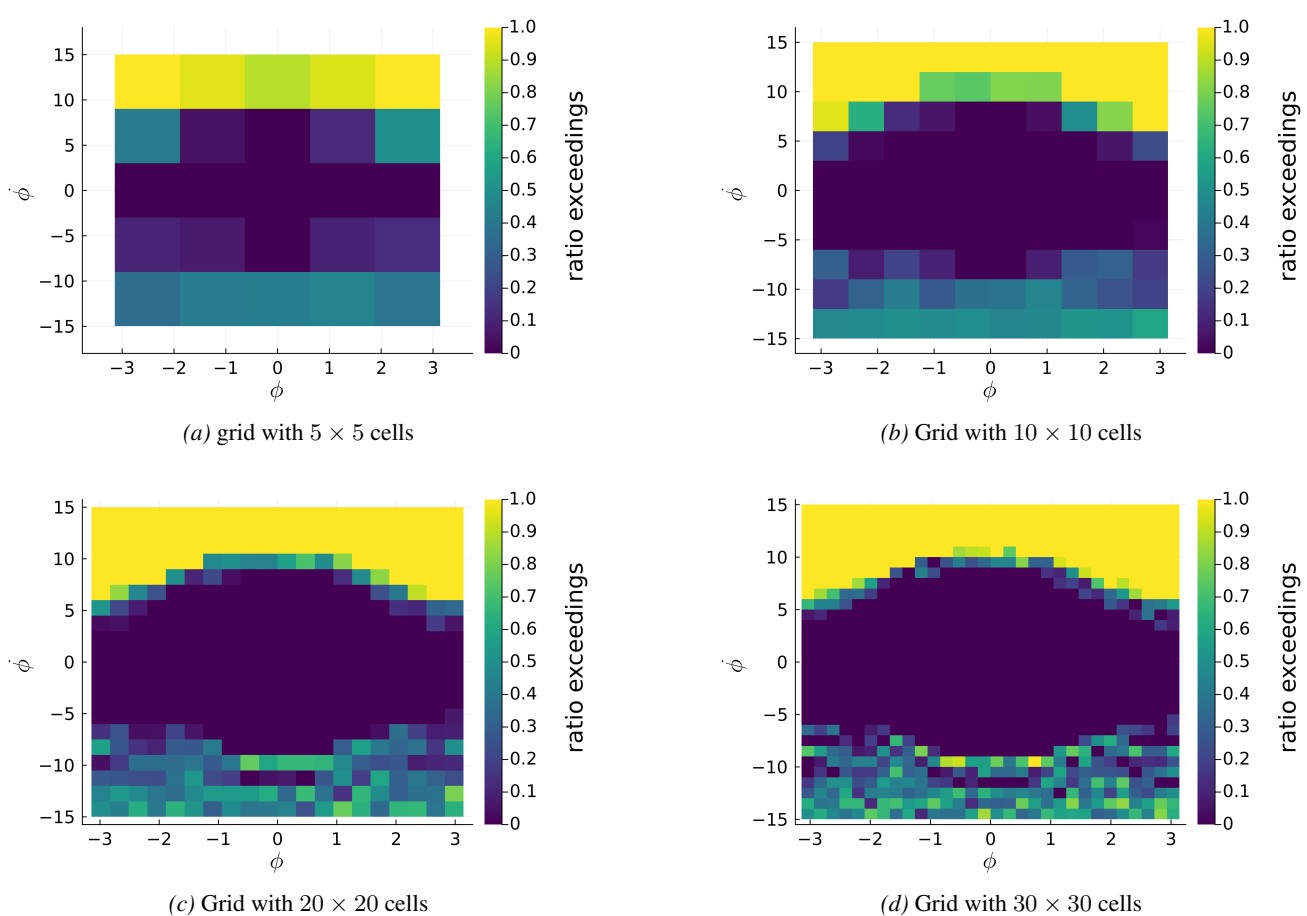

*(a)* grid with $5 \times 5$ cells

*(b)* Grid with $10 \times 10$ cells

*(c)* Grid with $20 \times 20$ cells

*(d)* Grid with $30 \times 30$ cells

*Figure 11.* Comparison of the grid size of the heatmaps with the following number per axis: 5, 10, 20, 30.

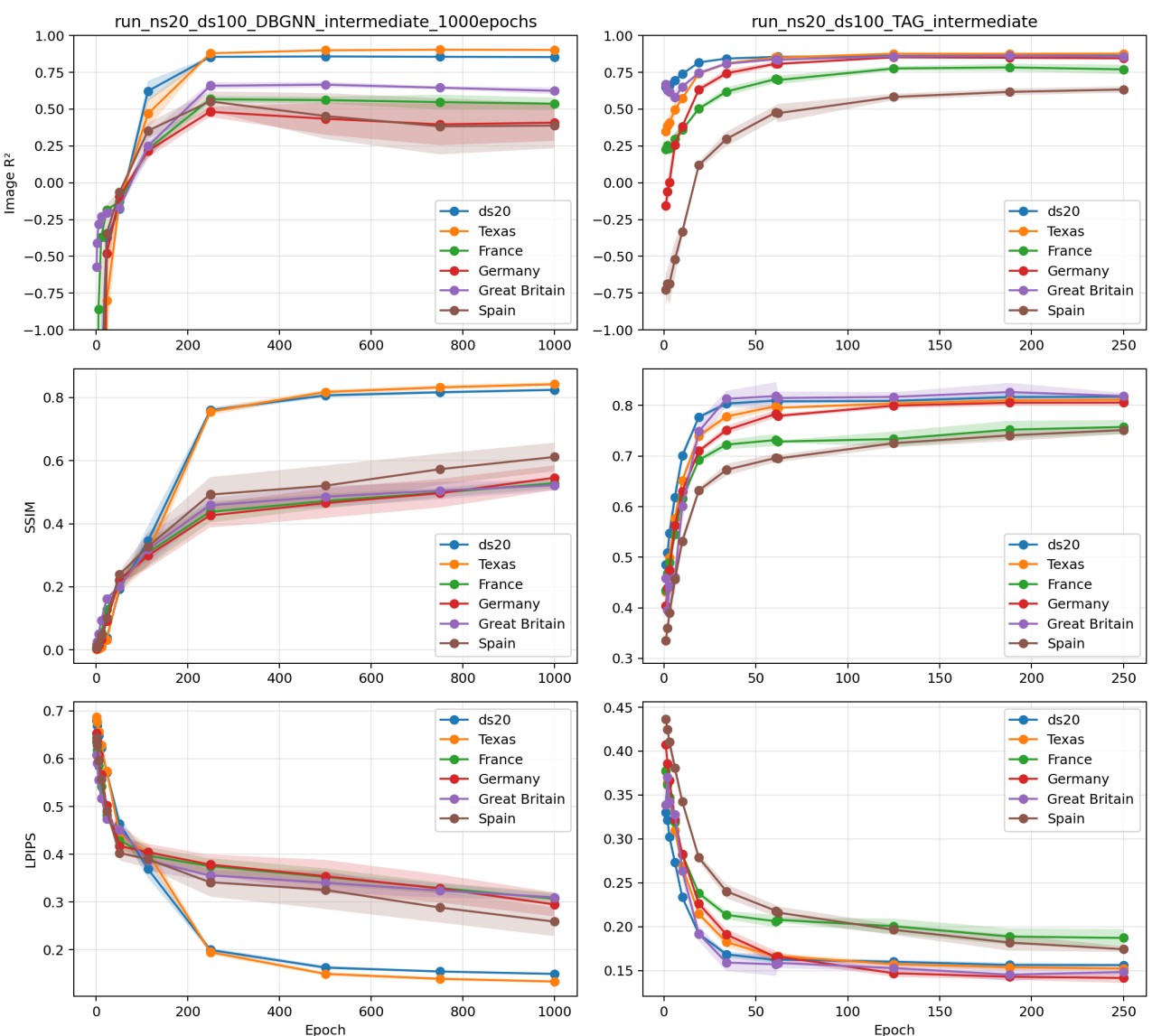

*Figure 12.* Training curves of DBGNN-MLP and TAG-MLP show the smoothness of TAG, which may show the strong out-of-distribution generalization.

*Table 13.* Out-of-distribution performance on the downstream task of predicting SNBS measured by $R^2$ in %.

| | Model | Germany | France | GB | Spain |
|---|---|---|---|---|---|
| MLP | TAG | $89.74_{\pm0.75}$ | $80.44_{\pm2.57}$ | $57.71_{\pm5.60}$ | $66.65_{\pm5.18}$ |
| | DBGNN | $43.73_{\pm8.23}$ | $44.00_{\pm7.94}$ | $< 0$ | $40.91_{\pm23.52}$ |
| CNN | TAG | $90.26_{\pm0.42}$ | $82.19_{\pm0.83}$ | $63.62_{\pm1.25}$ | $67.71_{\pm1.58}$ |
| | DBGNN | $30.71_{\pm7.33}$ | $35.13_{\pm5.83}$ | $< 0$ | $21.43_{\pm10.80}$ |

*Table 14.* Out-of-distribution performance on the downstream task of identifying critical contingencies, measured by intersection over union between predicted and true critical cells.

| Model | tr100evGermany | tr100evFrance | tr100evGB | tr100evSpain |
|---|---|---|---|---|
| TAG-MLP | $0.62_{\pm0.07}$ | $0.18_{\pm0.10}$ | $0.27_{\pm0.02}$ | $0.05_{\pm0.01}$ |
| DBGNN-MLP | $0.08_{\pm0.03}$ | $0.00_{\pm0.00}$ | $0.03_{\pm0.02}$ | $0.03_{\pm0.03}$ |
| TAG-CNN | $0.65_{\pm0.03}$ | $0.29_{\pm0.11}$ | $0.33_{\pm0.00}$ | $0.22_{\pm0.13}$ |
| DBGNN-CNN | $0.09_{\pm0.14}$ | $0.00_{\pm0.00}$ | $0.08_{\pm0.02}$ | $0.05_{\pm0.02}$ |

*Table 15.* Performance on predicting the relaxed down stream task of predicting critical contingencies, based on the proportion of cases in which all 20 most critical cells appear within the predictor's top 30.

| Model | tr20ev20 | tr100ev100 | tr20ev100 |
|---|---|---|---|
| TAG-MLP | $0.75_{\pm0.01}$ | $0.56_{\pm0.01}$ | $0.46_{\pm0.03}$ |
| DBGNN-MLP | $0.75_{\pm0.03}$ | $0.67_{\pm0.01}$ | $0.54_{\pm0.04}$ |
| TAG-CNN | $0.79_{\pm0.01}$ | $0.65_{\pm0.03}$ | $0.53_{\pm0.02}$ |
| DBGNN-CNN | $0.78_{\pm0.03}$ | $0.72_{\pm0.01}$ | $0.57_{\pm0.04}$ |

*Table 16.* Out-of-distribution performance on predicting the relaxed down stream task of predicting critical contingencies, based on the proportion of cases in which all 20 most critical cells appear within the predictor's top 30.

| Model | tr100evGermany | tr100evFrance | tr100evGB | tr100evSpain |
|---|---|---|---|---|
| TAG-MLP | $0.82_{\pm0.03}$ | $0.29_{\pm0.15}$ | $0.43_{\pm0.03}$ | $0.09_{\pm0.02}$ |
| DBGNN-MLP | $0.14_{\pm0.05}$ | $0.00_{\pm0.00}$ | $0.09_{\pm0.07}$ | $0.05_{\pm0.05}$ |
| TAG-CNN | $0.86_{\pm0.04}$ | $0.44_{\pm0.12}$ | $0.53_{\pm0.03}$ | $0.35_{\pm0.18}$ |
| DBGNN-CNN | $0.15_{\pm0.21}$ | $0.00_{\pm0.00}$ | $0.16_{\pm0.04}$ | $0.10_{\pm0.04}$ |

*Table 17.* Performance on predicting the heatmaps of the landscapes measured by SSIM in % with a grid size of $10 \times 10$.

| Model | In-Distribution | | Out-of-Distribution |
|---|---|---|---|
| | tr20ev20 | tr100ev100 | tr20ev100 |
| TAG-MLP-ns10 | $81.59_{\pm0.52}$ | $81.25_{\pm1.02}$ | $67.14_{\pm0.89}$ |
| TAG-MLP | $81.37_{\pm1.58}$ | $81.41_{\pm0.23}$ | $71.11_{\pm1.47}$ |

*Table 18.* Performance on predicting the heatmaps of the landscapes measured by LPIPS in % with a grid size of $10 \times 10$.

| Model | In-Distribution | | Out-of-Distribution |
|---|---|---|---|
| | tr20ev20 | tr100ev100 | tr20ev100 |
| TAG-MLP-ns10 | $0.10_{\pm0.00}$ | $0.10_{\pm0.00}$ | $0.18_{\pm0.00}$ |
| TAG-MLP | $0.16_{\pm0.00}$ | $0.15_{\pm0.00}$ | $0.22_{\pm0.00}$ |

*Table 19.* Performance on predicting the heatmaps of the landscapes measured by $R^2$ in % with a grid size of $10 \times 10$. For a grid size of $20 \times 20$, the full results are shown in Table 9.

| Model | In-Distribution | | Out-of-Distribution |
|---|---|---|---|
| | tr20ev20 | tr100ev100 | tr20ev100 |
| TAG-MLP-ns10 | $90.64_{\pm0.13}$ | $88.92_{\pm0.04}$ | $70.85_{\pm1.13}$ |
| TAG-MLP | $89.77_{\pm0.15}$ | $88.42_{\pm0.31}$ | $68.53_{\pm0.50}$ |

*Table 20.* Performance on the downstream task of predicting SNBS based on heatmap predictions measured by $R^2$ in % with a grid size of $10 \times 10$. For a grid size of $20 \times 20$, the full results are shown in Table 3.

| Model | In-Distribution | | Out-of-Distribution |
|---|---|---|---|
| | tr20ev20 | tr100ev100 | tr20ev100 |
| TAG-MLP-ns10 | $82.07_{\pm0.32}$ | $85.70_{\pm0.02}$ | $63.36_{\pm1.76}$ |
| TAG-MLP | $82.61_{\pm0.46}$ | $86.60_{\pm0.50}$ | $60.48_{\pm1.39}$ |

*Table 21.* Impact of grid resolution when identifying critical contingencies measured by IoU.

| Model | tr20ev20 | tr100ev100 | tr20ev100 |
|---|---|---|---|
| TAG-MLP-ns10 | $0.70_{\pm0.01}$ | $0.50_{\pm0.02}$ | $0.43_{\pm0.01}$ |
| TAG-MLP | $0.58_{\pm0.01}$ | $0.41_{\pm0.01}$ | $0.32_{\pm0.02}$ |

*Table 22.* Influence of number of training samples evaluated using SSIM in %.

| Model | tr20ev20 | tr100ev100 | tr20ev100 |
|---|---|---|---|
| TAG-MLP (1000 grids) | $76.89_{\pm0.76}$ | $80.53_{\pm0.38}$ | $65.42_{\pm1.22}$ |
| TAG-MLP (3000 grids) | $80.27_{\pm0.48}$ | $81.83_{\pm0.43}$ | $70.86_{\pm0.48}$ |
| TAG-MLP | $81.37_{\pm1.58}$ | $81.41_{\pm0.23}$ | $71.11_{\pm1.47}$ |

*Table 23.* Ablation studies varying the number of convolutional layers of TAG-MLP measured by SSIM in %.

| Model | layers | tr20ev20 | tr20ev100 | tr100ev100 |
|---|---|---|---|---|
| TAG-MLP | 3 | $78.74_{\pm0.49}$ | $78.12_{\pm0.40}$ | $68.63_{\pm0.44}$ |
| TAG-MLP | 5 | $81.37_{\pm1.58}$ | $81.41_{\pm0.23}$ | $71.11_{\pm1.47}$ |
| TAG-MLP | 8 | $82.16_{\pm0.65}$ | $82.30_{\pm0.36}$ | $72.93_{\pm0.63}$ |

