# OpenReview forum: "Learning Dynamic Stability Landscapes in Synchronization Networks"
_ICML.cc/2026/Conference — ICML 2026 regular_

### Official Review · Reviewer_rGBY · 2026-03-05

**Soundness:** 4
**Presentation:** 3
**Significance:** 3
**Originality:** 3
**Overall Recommendation:** 5
**Confidence:** 3

**Summary:**

An important concept is investigated by this manuscript—stability landscape prediction as a novel upstream task for assessing the robustness of synchronization in complex oscillator networks. The authors propose predicting full 2D stability landscape heatmaps (20×20 grids) per node from graph topology, extending prior work that predicted only scalar single-node basin stability (SNBS) values. Using a GNN encoder (TAG or DBGNN) with CNN/MLP decoders, the models predict these landscapes with approximately 85% SSIM in-distribution and 67-78% under size-shift generalization. The paper releases two datasets of 10,000 graphs each at 20 and 100 nodes, generated from second-order Kuramoto dynamics with ~500,000 CPU hours of simulation. The approach demonstrates zero-shot transfer to real power grid topologies (Germany, France, GB, Spain) with up to 80% SSIM, enabling downstream tasks like critical contingency screening.

**Compliance With Llm Reviewing Policy:**

Affirmed.

**Final Justification:**

I consider this paper to be a strong benchmark/task contribution: the move from scalar single-node basin stability prediction to full stability-landscape prediction is original, well motivated, and practically meaningful, and the accompanying dataset and evaluation setup provide clear value for future work. The method itself is technically reasonable, the experiments are broad for this type of paper, and the theoretical connection between heatmap error and downstream SNBS error supports the overall soundness. The rebuttal and follow-up reply addressed several of my main concerns constructively, including the threshold issue, runtime support, training-size effects, limitations, and a more practical contingency metric, which increased my confidence in the paper.

Some concerns remain only partially resolved: the paper still lacks a systematic cell-level uncertainty/calibration analysis, robustness beyond the current homogeneous Kuramoto setting is not fully explored, the TAG-vs-DBGNN gap on real topologies is still not explained mechanistically, and the architectural study could be more complete. However, for a benchmark/task paper, I view these as limitations that should be clarified rather than flaws that undermine acceptance. Overall, the rebuttal changed my evaluation positively, and I now believe the paper’s originality, significance, and overall technical solidity justify acceptance. I would encourage the authors to add further appendix material on these remaining points in the revised version.

**Key Questions For Authors:**

See the Weakness and the Limitations parts.

**Limitations:**

No dedicated sections for Related Work or Limitations are present. This gap in the manuscript structure impairs narrative integrity, leaving readers without clear guidance on the limitations and boundaries of the proposed contributions.

I would encourage the authors to add a concise discussion of some of these items: (i) dependence on the fixed Kuramoto setting and threshold choice, (ii) uncertainty introduced by Monte Carlo estimation and 20×20 discretization, (iii) the gap between synthetic training graphs and real-grid deployment, and (iv) the fact that predicted landscapes are best viewed as a screening tool before detailed physical simulation, not a direct operational substitute.

**Strengths And Weaknesses:**

## Strengths
1.
Original and well-motivated reformulation of the prediction target
The core idea—predicting a stability landscape $L_i^{(s)}(G,h)$ rather than only a scalar SNBS—is genuinely interesting, and the paper makes the relationship mathematically clean by showing how the scalar score is recovered as a mixture over regions. That is, in my view, the clearest source of originality in the submission. The paper does not present the landscape merely as a prettier visualization: it argues for concrete downstream utility, especially for identifying where instability occurs in perturbation space rather than only how much overall instability exists.

I also appreciate that the landscape formulation remains backward compatible with classical basin-stability quantities, rather than replacing them with a disconnected objective.The manuscript is reasonably transparent that the methodological novelty is not a brand-new GNN architecture per se, but a new task and benchmark with initial baselines. That honesty strengthens the paper’s framing rather than weakening it.

2.
Substantial benchmark contribution: The two synthetic datasets are nontrivial in scale ( 10,000 graphs each with per-node landscape labels), and the paper makes a convincing case that label generation is expensive enough to justify a learned surrogate, reporting roughly 500,000 CPU hours for dataset creation. The benchmark is thoughtfully designed around distribution shift: the 20-node and 100-node ensembles differ in target distribution, and the manuscript explicitly foregrounds the resulting OOD challenge rather than avoiding it.

Importantly, the benchmark is informative rather than merely convenient: models that look competitive on synthetic data separate sharply on real-topology transfer, especially TAG versus DBGNN. That is valuable because it tells us something substantive about generalization behavior, not just in-distribution fit.

3.
Solid technical grounding: The label construction is simple and auditable: each cell probability is defined from empirical counts $q_{i,m,n}$, and SNBS is recovered as a weighted sum over cells. That makes the supervision signal easy to understand and verify.

The theoretical component is stronger than I initially gave it credit for. Theorem A.1 is not just an intuition; Appendix A.2 provides a rigorous proof using Jensen’s inequality, and the result gives a meaningful guarantee that weighted pixel-wise error upper-bounds downstream SNBS error.The choice of long-range-capable encoders is also well motivated in the paper itself: Sec. 2.2/3.1 explicitly ties the task to non-local behavior and cites prior work on long-range dependencies in dynamic stability and power-grid settings, including the cited SNBS prediction literature and Ringsquandl et al. (2021).

I found the decision to use lightweight MLP/CNN decoders reasonable for a first benchmark paper. The manuscript explicitly says more expressive generative decoders are deferred to future work to keep the initial comparison simple and interpretable (Sec. 3.1).

4.
The empirical evaluation is broad and informative: the paper reports SSIM, $R^2$, LPIPS, downstream SNBS prediction, real-topology transfer, and contingency-screening performance rather than relying on a single metric.The reported synthetic results are genuinely strong for a first pass on this task: the best SSIM reaches 85.89% in-distribution and 78.50% under the 20→100 size shift.On the downstream task, the paper shows that predicting the richer object does not destroy usefulness for classical scalar stability prediction; the best heatmap-based SNBS $R^2$ remains competitive, reaching 90.08% in-distribution and 73.73% OOD. The real-topology results are uneven but still meaningful. In particular, TAG-MLP and TAG-CNN show that zero-shot transfer to much larger realistic grids is not hopeless, which is an important result for the significance of the benchmark.

5.
Helpful presentation and reproducibility effort

Figure 2 is useful because it makes the discretization pipeline concrete by contrasting raw Monte Carlo samples with the derived heatmaps, which helps the reader understand what is being predicted.Figures 4, 7, and 8 provide qualitative examples that are worth having; they make it easier to interpret what the summary metrics are actually measuring and support the claim that large-scale structure is often captured well.

The Software and Data section also signals good intent toward reproducibility by promising code/data release and providing sample review-time code with replication instructions.

## Weaknesses

1.
Important modeling assumptions are only partially stress-tested: The experimental setting is fairly specific: a second-order Kuramoto model with fixed $\alpha = 0.1$, homogeneous coupling $K=9$, binary node feature $P_i\in ${−1,+1}, and a fixed perturbation box $(\phi,\dot \phi)\in [−π,π]×[−15,15]$. This is a reasonable starting point, but it limits how broadly one can read the empirical conclusions. The stability label itself depends on the threshold $|\dot \phi| \leq 0.1$, adopted from prior work.  Likewise, perturbations are sampled uniformly from a fixed box. No direct evidence found in the paper for threshold-sensitivity experiments around that choice, even though it defines the target labels, and for whether the learned landscapes remain stable under different or more realistic perturbation distributions.

The paper defines cell counts $q_{i,m,n}$ and acknowledges sampling-noise/statistical-reliability tradeoffs, but it does not quantify uncertainty with confidence intervals or calibration at the cell level.

2.
The grid-resolution discussion is useful, but the downstream implications of discretization are still not fully nailed down.

To be clear, Appendix A.8 is a real positive: it compares 5×5, 10×10, 20×20, and 30×30 grids and argues that 20×20 is a good compromise between spatial detail and statistical reliability. The reported comparison also supports that 10×10 is weaker than 20×20 in the provided study, so the choice is not arbitrary.

What I still miss is a more direct analysis of how discretization affects decision-oriented use cases, especially contingency screening, beyond broad heatmap metrics. In other words, the representation study is helpful, but the practical consequences of the chosen resolution could be discussed more explicitly.

No direct evidence found in the manuscript for a calibration-style analysis of whether finer or coarser grids change the reliability of high-risk-cell identification. Given the intended use case, that would strengthen the practical argument.

3.
The notation in Eq. (7), written as $|\dot\phi (\delta)|\leq 0.1$ with $\delta = (\phi, \dot \phi)$, is slightly awkward but understandable in context. I would still rewrite it more explicitly in a final version for readability (Eq. (7)).

4.
The benchmark reveals interesting transfer behavior, but the error analysis is still not deep enough.

The most striking example is the TAG–DBGNN gap on real topologies. On Table 4 and Tables 9–11, TAG-based models are dramatically more stable than DBGNN-based ones on real grids, despite much closer performance on synthetic settings. That is exactly the kind of result that makes the benchmark valuable—but the paper does not yet explain why it happens.

Authors themself note an important nuance: a prediction can capture the shape of the stable region while still missing the exact exceedance ratios or final SNBS value. That observation is insightful, and it also suggests that additional metrics such as boundary localization quality or calibration of unstable-region probabilities would be very useful (Sec. 4.1).

The contingency-screening evaluation is interesting, but the IoU numbers are only moderate in-distribution and often weak under OOD/real-topology settings. Because Appendix A.7 uses an intentionally strict “exact same cell” criterion, a near-miss is treated the same as a poor prediction; I would like to see a complementary tolerance-aware or ranking-based analysis here. Besides, the appendix gives one possible explanation for weak downstream performance on Great Britain—its unique dynamical properties—but overall the manuscript still feels light on topology-specific failure analysis.

5.
The paper does some things well here: it reports training times on an A100, summarizes architecture/training settings, and provides sample code plus a release plan. However, the conclusion claims evaluation is reduced to “seconds per graph,” and I could not find a controlled per-graph inference benchmark or runtime table supporting that statement directly.  How does prediction accuracy scale with the amount of training data and time? Could the model achieve comparable performance with fewer training graphs or fewer perturbations per node during training?

6.
Insufficient Architectural Ablation Studies: No ablation studies are provided for architectural components beyond the grid resolution investigation. Key questions remain unanswered: How does performance vary with number of GNN layers? What is the impact of hidden dimension choices (1751 for TAG vs 512 for DBGNN)?

The choice between TAG and DBGNN is not systematically analyzed. While both are used, the paper does not identify which architectural properties (local vs long-range aggregation, polynomial filters vs micro-propagation steps) contribute most to performance.
No comparison to simpler baselines (e.g., MLP on node features alone, GCN) is provided, making it difficult to assess whether the sophisticated architectures are necessary for this task.

---

> ### Author Rebuttal · Authors · 2026-03-30
>
> We have added a dedicated Limitations section; please see our response to Reviewer bdh2 for details.
>
> **Answers for W1:**
>
> There is essentially no sensitivity to this threshold. Asymptotically the system either synchronizes or runs towards a limit cycle with phase velocity around 10/s. Earlier studies of re-synchronization explored sophisticated heuristics to robustly identify the cases where we have re-synchronization, but experience shows that the simple threshold 0.1 is sufficient and achieves identical probabilistic results in the setting explored here. We added histograms of the maximum final frequency deviation and indicated the 0.1 threshold (<https://anonymous.4open.science/r/SNBS_Histograms-642B/README.md>, max_final_freq_hist.png). Almost no configurations lie just above this cutoff. Because the same threshold was also used as an early-stopping criterion for synchronization, values on the synchronized side of the boundary cannot be resolved in finer detail. This is, however, consistent with current state-of-the-art practice in oscillator-network studies.
>
> Regarding the homogeneous parameter settings, we added a limitations section that discusses this issue (see Reply to Reviewer bdh2).
>
> **Answers for W2:** Below, we report IoU results for the downstream critical-contingency identification task using TAG-MLP trained at **20×20** and **10×10** resolutions. As expected, performance improves at lower resolution (fewer cells).
>
> | Model | tr20ev20 | tr20ev100 |
> |-------|----------|-----------|
> | TAG-MLP (ns20) | 0\.58 | 0\.32 |
> | TAG-MLP (ns10) | 0\.70 | 0\.43 |
>
> **Answers for W3:** We thank the Reviewer for pointing this out. In the revised manuscript, we rewrote this Equation explicitly in terms of the maximum absolute final frequency over all nodes after applying perturbation $\\boldsymbol{\\delta}$, and introduced a separate symbol for this quantity, $\\omega^{\\mathrm{final}}\_j(\\boldsymbol{\\delta})$, to avoid overloading $\\dot{\\phi}$. We also added a short sentence after the equation clarifying this definition.
>
> **Answers for W4:**
>
> See reply to Reviewer to Reviewer  SPxZ (Answers for W2, W3, W8, Q2, and Q4) and Reply to Reviewer bdh2: Limitations.
>
> We agree that an exact cell-by-cell criterion is stringent, and that more operationally relevant metrics should be considered in practice. Here, we include this task primarily to demonstrate the potential of deriving meaningful downstream metrics from predicted landscapes. We would be happy to evaluate alternative metrics if the reviewer has specific suggestions. In addition, the effect of this strict criterion can be mitigated at lower resolutions, as illustrated by our critical-cell prediction at 10×10 resolution (see Answer to W2).
>
> **Answers for W5:**  The inference table below reports the runtime for evaluating the full test sets (1,500 grids), showing that more than 1,000 grids can be analyzed in under one second. This demonstrates the huge advantage over running Monte Carlo simulations requiring several hours per grid.
>
> | Model | Runtime ds20 test split \[s\] | Runtime ds100 test split \[s\] |
> |-------|-----------------------------|------------------------------|
> | TAG-MLP | 0\.25 | 0\.34 |
> | TAG-CNN | 0\.26 | 0\.40 |
>
> The table below reports how performance (SSIM, %) varies with the number of training grids. Increasing the number of grids improves accuracy, while the method remains effective even with fewer training grids.
>
> | Model | tr20ev20 | tr20ev100 |
> |-------|----------|-----------|
> | TAG-MLP (1000 grids) | 76\.89 | 65\.42 |
> | TAG-MLP (3000 grids) | 80\.27 | 70\.86 |
> | TAG-MLP (all grids) | 81\.32 | 81\.61 |
>
> In the literature, prior work on basin-stability prediction (Fig. 4 in <https://arxiv.org/pdf/2402.17500>) shows that GNN performance generally improves with larger training sets, while still remaining viable with fewer samples. That study also suggests that the key driver is the number of training nodes rather than the number of distinct grids. In this sense, our comparison between dataset20 and dataset100 provides an implicit test of training-set size effects: dataset100 contains roughly five times more training nodes. This difference is likely an important contributor to the performance gap between tr100ev100 and tr20ev20.
>
> **Answers for W6:**
> We conducted ablation studies on the number of TAG layers for dataset20 (SSIM in %), indicating the advantage of more layers.
>
> | Model | \# GNN layers | tr20ev20 | tr20ev100 |
> |-------|--------------|----------|-----------|
> | TAG-MLP | 3 | 79\.30 | 69\.48 |
> | TAG-MLP | 5 | 81\.32 | 81\.61 |
> | TAG-MLP | 8 | 82\.60 | 73\.37 |
>
> Additional ablations on dataset100 will be included in the camera-ready version.
>
> Regarding the use of DBGNN and TAG, we followed prior work (10.1063/5.0160915), which showed for SNBS prediction that architectures with stronger long-range propagation (such as TAG) outperform standard GCNs. We therefore focused on these models.

---

> > ### Author Rebuttal · Reviewer_rGBY · 2026-04-02
> >
> > Thank you for the detailed rebuttal. It resolved several of my specific concerns: the Eq. (7) notation will be rewritten more clearly, the added runtime table now substantiates the fast-inference claim, the additional training-set-size and layer-ablation results are helpful, and the dedicated limitations discussion appropriately narrows the scope of the empirical claims. The newly provided maximum-final-frequency histogram is also useful evidence that the specific `|\dot\phi| <= 0.1` threshold is not materially driving the reported statistics in the current setting. The added `10x10` contingency results and the SNBS histogram discussion further help clarify why the real-topology transfer setting should be interpreted as a deliberately hard feasibility check rather than as a clean benchmark.
> >
> > I still have a few follow-up questions before I would consider my concerns fully resolved. First, beyond the now-better-justified `0.1` threshold, can you summarize whether robustness has been checked for alternative perturbation distributions or more heterogeneous settings, and whether any uncertainty / calibration analysis is available for the cell-level probabilities? Second, beyond general distribution shift, what do you believe is the main reason for the very large TAG-vs-DBGNN gap on the real topologies? Third, for contingency screening, would you consider reporting a tolerance-aware or ranking-based metric in addition to exact-cell IoU, since that seems closer to the intended use case? Overall, now I keep my current score for further discussion, but since the given detailed rebuttal and the additional experiments, I am open to raising it.

---

> > > ### Author Response · Authors · 2026-04-06
> > >
> > > We thank the Reviewer for the detailed follow-up questions.
> > >
> > >
> > > **Regarding 1)** Each node receives a different set of random perturbations, resulting in varying perturbation distributions across cells (including variation in the number of perturbations per cell). While our dataset contains several different perturbation distributions, we did not systematically study distinct sets for identical nodes. The uncertainties per cell are substantial due to the relatively sparse sampling—on average, only 25 perturbations per cell for 20×20 image resolution.  As an illustration, in a Bernoulli setting with $p=0.5$  and $n=25$ , the empirical proportion has standard deviation $\sqrt{p(1−p)/n}=0.1$. In particular, cells with p close to 0 or 1 will have almost no dependence on the within-cell distribution. The location of the boundary between stable and unstable regions will not change meaningfully.
> > >
> > > Overall, when predicting across a large set of cells spanning multiple nodes and grids, the inherent sparsity can be viewed as noise, which the machine learning models appear to handle effectively.
> > >
> > > **Regarding 2)** We have added new results and provided further clarification (<https://anonymous.4open.science/r/SNBS_Histograms-642B/README.md>, ns20_ds100_dbg_vs_tag_metrics_grid.png). Please refer to our second reply to Reviewer SPxZ.
> > >
> > > **Regarding 3)** We introduce a relaxed evaluation metric for assessing performance in identifying the most critical cells. Specifically, we compute accuracy by evaluating whether the 20 most critical cells according to the ground truth labels fall within the 30 most critical cells according to the model predictions. This approach softens the original strict requirement while remaining practically relevant, as it tests whether the model can identify the most critical cells within a reasonably constrained set.
> > >
> > > The following table below reports the proportion of cases in which all 20 most critical cells appear within the predictor’s top 30.
> > > | Model | tr20ev20 | tr20ev100 | tr100ev100 | tr100evFrance | tr100evGB | tr100evGermany | tr100evSpain | tr100evTexas |
> > > |-------|----------|-----------|------------|---------------|-----------|----------------|--------------|--------------|
> > > | TAG-MLP | 0\.754 | 0\.461 | 0\.555 | 0\.290 | 0\.430 | 0\.820 | 0\.090 | 0\.555 |
> > > | DBGNN-MLP | 0\.754 | 0\.536 | 0\.673 | 0\.000 | 0\.090 | 0\.140 | 0\.050 | 0\.673 |
> > > | TAG-CNN | 0\.790 | 0\.529 | 0\.650 | 0\.440 | 0\.530 | 0\.860 | 0\.350 | 0\.650 |
> > > | DBGNN-CNN | 0\.775 | 0\.569 | 0\.719 | 0\.000 | 0\.160 | 0\.150 | 0\.100 | 0\.719 |
> > >
> > > We hope these new results help address the concerns and clarify the strengths of our work.

---

### Official Review · Reviewer_SPxZ · 2026-03-11

**Soundness:** 3
**Presentation:** 2
**Significance:** 3
**Originality:** 3
**Overall Recommendation:** 4
**Confidence:** 3

**Summary:**

This paper introduces a new ML task: predicting per-node basin stability landscapes for power grid synchronization networks. Rather than predicting a single scalar stability score (SNBS) per node — as prior work did — the model predicts a full 20×20 heatmap over the 2D perturbation space (phase angle φ, frequency deviation φ̇), showing which fault conditions will cause the grid to lose synchronization. Two large datasets are released (10,000 graphs each at 20 and 100 nodes), generated via 500,000 CPU hours of Monte Carlo simulation. A GNN encoder (TAG or DBGNN) produces node embeddings, and an MLP or CNN decoder outputs the heatmap per designated node. The best model achieves 85% SSIM in-distribution, 78% under 20→100 node size shift, and transfers zero-shot to real power grids (Germany, France, GB, Spain) at up to 83% SSIM, reducing evaluation from thousands of CPU hours to seconds per graph.

**Compliance With Llm Reviewing Policy:**

Affirmed.

**Final Justification:**

I am increasing my score following the author's rebuttal and follow-up response. The authors provided a clear technical explanation for the performance discrepancy between synthetic and real-world grids, supported by new training curve analysis. Specifically, they demonstrated that while DBGNN achieves higher peak performance on in-distribution data, its optimization landscape is less "smooth" than TAG’s, leading to slower convergence and limited out-of-distribution generalization. The authors' transparency regarding these training dynamics—and their commitment to including these insights and potential regularization strategies in the final version—sufficiently addresses my concerns regarding the model's robustness and architectural trade-offs.

**Key Questions For Authors:**

1. what random graph model was used to generate the 10,000 graphs in each ensemble? The topology distribution of synthetic graphs directly determines what the GNN learns and how well it transfers to real grids. Without this information the domain gap between synthetic and real grids cannot be analyzed or addressed.
2. DBGNN consistently outperforms TAG on synthetic data but collapses completely on real grids (44-54% SSIM, high variance). What causes this reversal? Is it overfitting to synthetic topology patterns, sensitivity to degree distribution differences, or something else?
3. what ODE solver and simulation time horizon were used for dataset generation? How was the synchronous steady state found for each graph?
4. TAG-MLP IoU on real grids ranges from 6% (Spain) to 57% (Germany). What drives this inconsistency across grids? Is it related to grid size, topology, or SNBS distribution? A model with 6% IoU on contingency identification is not operationally useful?
5. The landscape covers φ ∈ [-π,π] and φ̇ ∈ [-15,15]. How were these ranges chosen? Do all practically relevant fault conditions fall within this space for real grids, or are there realistic fault scenarios that fall outside these bounds and would be missed by the landscape?

**Limitations:**

Yes

**Strengths And Weaknesses:**

Strengths
1. The landscape strictly subsumes scalar SNBS, enabling downstream tasks like contingency screening that scalar prediction cannot support
2. 500,000 CPU hours of simulation compressed into a reusable benchmark. GNN inference reduces this to seconds per graph
3. TAG-MLP achieves 77-83% SSIM on real power grids without retraining, demonstrating practical applicability beyond synthetic data
4. Changing the perturbation distribution ρ(p) requires only reweighting existing landscape cells, not regenerating the dataset or retraining

Weaknesses
1. The paper introduces three nested mathematical abstractions (s, SNBS, L) before showing a concrete example. Figure 1 should appear on page 1 with a plain explanation of what a stability landscape is, before any equations. A reader unfamiliar with power grids would struggle through two pages of formalism without understanding what problem is being solved.
2. Despite outperforming TAG on synthetic data, DBGNN achieves only 44-54% SSIM with high variance on real grids. The paper does not explain why, and the synthetic graph generation process is not described, making the domain gap between synthetic and real grids impossible to analyze.
3. TAG-MLP IoU ranges from 6% (Spain) to 57% (Germany) across real grids, too inconsistent for practical deployment. The strict exact-cell evaluation metric also makes results hard to interpret operationally
4. The framework only handles faults at one node at a time. Real grid failures often involve simultaneous or cascading faults across multiple nodes or edges, which this approach cannot address.
5. Equations 2 and 5 are essentially identical. SNBS with general s versus basin stability BS. Same for Equations 3 and 5 (LBS). These could be collapsed with a single sentence clarifying that BS is the specific stability criterion used. The repetition adds length without adding clarity.
6. ρ(p) and ρ(p|h) never explained intuitively. They appear as measure-theoretic notation without ever saying plainly:
ρ(p)   = sample perturbations uniformly across full (φ, φ̇) space
ρ(p|h) = sample only within one cell of the 20×20 grid
A one-line intuitive explanation would make these immediately accessible to readers outside the dynamical systems community.
7. Section 2 describes what the dataset contains but not how the synthetic graphs were generated — what random graph model was used, what ODE solver was used, what simulation time horizon was chosen, or what HPC infrastructure was used. These are essential for reproducibility and understanding the domain gap between synthetic and real grids.
8. Table 4 shows DBGNN collapsing completely on real grids but the paper provides no analysis or explanation. This deserves at least a paragraph of discussion given it directly contradicts the synthetic data results and has significant practical implications.

---

> ### Author Rebuttal · Authors · 2026-03-30
>
> **Answers for W1:**  We changed the third paragraph of the introduction to introduce the landscape plots as the motivation.
>
> [...] Before introducing the technical framework, we illustrate the motivation using the stability landscape plots in Figure 1. The axes represent perturbation magnitudes: points near the origin correspond to small perturbations, while points farther away correspond to larger perturbations. Dark purple indicates stable outcomes, whereas lighter green and yellow indicate instability. [..]
>
> **Answers for W2, W3, W8, Q2, and Q4:**
>
> The real-topology experiments should be viewed as a **feasibility check** rather than a statistically robust benchmark, because several real grids contain only about 100 nodes; the more statistically reliable transfer test is **dataset20 → dataset100**. In addition, the synthetic training graphs were not designed to match any specific real topology, so exact transfer is not expected (e.g., Texan-like training data are not naturally optimal for Spain). As shown by the SNBS histograms (<https://anonymous.4open.science/r/SNBS_Histograms-642B/README.md>, `SNBS_Histograms.png`), synthetic and real networks differ structurally, making this a deliberately hard transfer setting; the observed performance therefore indicates strong generalization.
>
> In the appendix, we added “Details on the dataset generation,” which describes how the synthetic topologies are constructed. Specifically, the synthetic networks are generated with a random-growth algorithm (<https://doi.org/10.1140/epjst/e2014-02279-6>) designed to reproduce key topological properties of power grids, including a realistic degree distribution.
>
> **Answers for W3:** The authors agree with the Reviewer regarding the limitation of the exact-cell evaluation and believe more sophisticated metrics should be used for operation. The task is only added to show the potential of deriving metrics from the landscapes. We are happy to evaluate other metrics, if the reviewer has an idea.
>
> **Answers for W4:**  In principle, the ML framework can be extended to multi-node faults. The main limitation is computational cost: defining multi-node stability robustly requires far more simulations than the single-node case. Instead of running 10,000 perturbations per node independently, one must sample a much larger space of joint perturbations affecting several nodes at once.
>
> When it comes to cascading failures, these are typically triggered by a single initial fault. The perturbations classified as unstable in the basin stability sense are indeed those that would trigger further failures and initiate a cascade. Including a detailed analysis of the resulting cascade, and understanding whether it stops or grows into a complete system collapse would require considerably more sophisticated models than used here and is out of scope.
>
> **Answers for W5 and W6:** We thank the Reviewer for this helpful comment. In the original version, Eqs. (2) and (3) introduced a general probabilistic stability framework, which we then specialized to basin stability in Eq. (5). We agree that this added unnecessary abstraction and redundancy.
>
> In the revised manuscript, we therefore removed the general notation based on $s$, $\\mathrm{SN}^{(s)}$, and $\\mathrm{L}^{(s)}$, and now introduce the problem directly in terms of $\\mathrm{BS}$, $\\mathrm{SNBS}$, and $\\mathrm{LBS}$. This collapses the repeated equations into a single concrete formulation and makes the introduction more direct.
>
> To better illustrate the intuition behind $\\rho(p)$ and $\\rho(p\\mid h)$ was not stated clearly enough. In the revised manuscript, we now explain this explicitly in plain language: $\\rho(p)$ denotes sampling perturbations over the full perturbation space, while $\\rho(p\\mid h)$ denotes sampling restricted to a specific region or grid cell $h$. We also tie this directly to our concrete setting $p=(\\phi,\\dot{\\phi})$.
>
>
> **Answers for W7, Q1, and Q3:** We added a subsection in the appendix with details on the numerical simulations. The differential equations were solved using DifferentialEquations.jl (10.5334/jors.151), with Tsit5 as the numerical integrator(10.1016/j.camwa.2011.06.002).
>
> The synchronous static state is computed by solving the nonlinear nodal balance equations, where node phases are mapped to complex phasors and network flows are derived from the Laplacian. The root finder `nlsolve`  from the NLSolve.jl package starts from a pseudoinverse-based initial guess.
>
> **Answers for Q5:**
>
> The phase range covers the full state space, the frequency range covers the attractor the generators move towards when they desynchronize, which is roughly at omega = 10. We believe the range is rather too large than too small as a frequency perturbation that reaches the frequency the generator achieves when desynchronizing would in practice already trigger protection mechanisms.

---

> > ### Author Rebuttal · Reviewer_SPxZ · 2026-04-02
> >
> > I appreciate the authors' response, but my concern regarding Weakness 2 was not directly addressed. Specifically, I am looking for an explanation as to why DBGNN achieves only 44–54% SSIM on real-world grids—performing significantly worse than TAG—despite its superior performance on synthetic data. This discrepancy suggests a potential lack of robustness or generalization that the rebuttal did not clarify.

---

> > > ### Author Response · Authors · 2026-04-06
> > >
> > > We thank the reviewer for the follow-up question. To address the raised concern, we have added new results and provide further clarification below.
> > >
> > > Specifically, to analyze the performance differences between TAG and DBGNN, we report training performance across epochs and provide the uploaded figure (https://anonymous.4open.science/r/SNBS_Histograms-642B/README.md, ns20_ds100_dbg_vs_tag_metrics_grid.png), which shows the first 250 epochs. At this stage, DBGNN has not yet fully converged, which explains the small gap compared with the table results. We will include fully converged plots in the camera-ready version. The curves show that TAG training is much smoother, reaching reasonable performance within a few steps, whereas DBGNN training is more difficult early on (including negative Image $R2$ in the first epochs). Later, in-distribution performance improves rapidly, while generalization to real-world topologies saturates. This suggests that TAG is smoother overall, and smoother models are often associated with better generalization. To improve out-of-distribution generalization for DBGNN, additional regularization may help, and we are happy to investigate this for the camera-ready version. Finally, we emphasize that our main generalization target is tr20ev100; real-world topologies are treated as an additional indicator of model generalization.
> > > We hope these additions help to resolve the concern.

---

### Official Review · Reviewer_bdh2 · 2026-03-13

**Soundness:** 4
**Presentation:** 4
**Significance:** 4
**Originality:** 3
**Overall Recommendation:** 6
**Confidence:** 5

**Summary:**

The authors lay out a full, novel computational pipeline, from conceptual premises to data set generation to a first baseline model, for the problem of predicting the behavior of coupled oscillators under perturbation. They focus on the interesting case of understanding how individual nodes behave qualitatively (whether or not they remain synchronized with respect to a rotating reference frame) under the influence of perturbations to their phase and frequency. Instead of predicting the effect of a single perturbation, they investigate the more general problem of inferring an entire landscape of perturbation effects for all given nodes in different graph topologies. After generating a rather large data set of such landscapes, they propose a graph neural network architecture which performs well on the task and can generalize to real topologies taken from electric power grids. Finally, they identify “critical contingencies” from their landscapes, regions where small changes lead to large system outcomes.

**Compliance With Llm Reviewing Policy:**

Affirmed.

**Key Questions For Authors:**

I have no questions which could raise my score.

**Limitations:**

The authors do not list limitations. In that sense, I think they should discuss at greater length whether or not their results generalize beyond the oscillators studied here.

**Strengths And Weaknesses:**

Soundness: Solid motivation and experimentation. Well-written and with good validation and benchmarking.

Presentation: Very nice-looking paper and the logical flow is clear. I was a little confused by the nature of the second downstream task which “[predicts] the volume of the basin stability, a metric derived from the heatmaps.” I believe these are the results shown in Table 3, but it is not immediately clear from the text.

Significance: The concrete products of this study in the form of the dataset of stability landscapes and the baseline GNN encoder are useful and will help the community study these synchrony phenomena in the future.

Originality: To my knowledge, the notion of stability landscape introduced here is novel.

---

> ### Author Rebuttal · Authors · 2026-03-30
>
> The authors would like to thank the reviewer for their very positive comments and the score.
>
> **Answers for W1:**  Yes, the Reviewer is correct. The second downstream task is simply the recovery of the scalar single-node basin stability (SNBS) from the predicted heatmaps, and these results are indeed those reported in Table 3.
>
> In the revised manuscript, we replaced the ambiguous phrase “predicting the volume of the basin stability” with a more direct description, namely predicting SNBS from the predicted heatmaps. We also revised the surrounding text to explicitly state that Table 3 reports this downstream scalar evaluation.
>
> > Limitations: The authors do not list limitations. In that sense, I think they should discuss at greater length whether or not their results generalize beyond the oscillators studied here.
>
> **Answers:** We added a section collecting all limitations, right before the Conclusion in the draft and show it below:
>
> ### Limitations
>
> Our work is subject to several important limitations that should be considered when interpreting the results. First, we focus on homogeneous Kuramoto oscillator networks with uniform coupling and parameters. While these systems are of theoretical interest and provide a foundation for understanding synchronization dynamics, they do not directly represent real-world power grids or other practical applications. The underlying ensembles are inspired by power grid topologies but incorporate simplifying assumptions that limit direct applicability. Nevertheless, the synchronization problem remains fundamentally relevant, and our landscape-based analysis has broader significance for the physics community.
>
> Second, our pipeline is evaluated on a single oscillator model with homogeneous parameters. Extending the analysis to richer dynamical systems and heterogeneous network properties would strengthen the generalizability of our approach. Recent work showed that similar stability tasks as considered here can successfully be addressed in the presence of heterogeneous node and edge features with different dynamical actors in more realistic power grid models [R1]. Demonstrating that this type of realistic model allows for rich landscape-like tasks is left for future work.
>
> Third, some of the real power grids analyzed in this work feature small numbers of nodes, which limits the statistical power for evaluating model generalization through direct benchmarking across different models. Nevertheless, demonstrating any generalization to such constrained systems provides compelling evidence that machine learning models can learn underlying physical principles across different networks. Additionally, more sophisticated machine learning decoding architectures could further enhance predictive performance beyond the methods presented here.
>
> Despite these limitations, we believe this work represents an important contribution to the field. The primary bottleneck of our approach is data scarcity—to our knowledge, no publicly available datasets currently exist for validating this pipeline. We hope this work serves as a proof of concept, encouraging the research community to invest in generating large-scale datasets across related domains.
>
> [R1] Nauck, Christian, et al. "Predicting the Fault‐Ride‐Through Probability of Inverter‐Dominated Power Grids Using Machine Learning." *IET Generation, Transmission & Distribution* 20.1 (2026): e70264.

---

> > ### Author Rebuttal · Reviewer_bdh2 · 2026-04-03
> >
> > I thank the authors for their reply and for their planned adjustments in the limitations section. I maintain my score.

---

### Decision · Program_Chairs · 2026-04-30

**Decision:**

Accept (regular)

**Comment:**

This paper introduces a  machine learning framework for predicting per-node stability landscapes in coupled oscillator networks, particularly power grids. Instead of a single scalar stability score, the model outputs a full 2D heatmap over perturbations, enabling richer analysis of synchronization behavior and identification of critical contingencies. The paper also contributes a large-scale dataset generated via extensive simulations and a GNN model that achieves strong performance and generalizes to real-world grid topologies.

The reviewers agree the work is technically solid, well-motivated, and impactful, highlighting the originality of the formulation, the value of the dataset, and the good empirical results. Some concerns related to the clarity and mathematical abstraction, missing details on dataset generation, and unexplained performance gaps across models and real-world settings have been adequately addressed during the response period. Overall, the paper makes an interesting contribution, and the reviewers are overall positive toward acceptance.